# Dithiine-linked metalphthalocyanine framework with undulated layers for highly efficient and stable $H_2O_2$ electroproduction

Qianjun Zhi[1], Rong Jiang[1], Xiya Yang[1], Yucheng Jin[1], Dongdong Qi[1], Kang Wang[1] ✉, Yunpeng Liu[2] ✉ & Jianzhuang Jiang[1] ✉

Realization of stable and industrial-level $H_2O_2$ electroproduction still faces great challenge due large partly to the easy decomposition of $H_2O_2$. Herein, a two-dimensional dithiine-linked phthalocyaninato cobalt (CoPc)-based covalent organic framework (COF), CoPc-S-COF, was afforded from the reaction of hexadecafluorophthalocyaninato cobalt (II) with 1,2,4,5-benzenetetrathiol. Introduction of the sulfur atoms with large atomic radius and two lone-pairs of electrons in the C-S-C linking unit leads to an undulated layered structure and an increased electron density of the Co center for CoPc-S-COF according to a series of experiments in combination with theoretical calculations. The former structural effect allows the exposition of more Co sites to enhance the COF catalytic performance, while the latter electronic effect activates the 2e$^-$ oxygen reduction reaction (2e$^-$ ORR) but deactivates the $H_2O_2$ decomposition capability of the same Co center, as a total result enabling CoPc-S-COF to display good electrocatalytic $H_2O_2$ production performance with a remarkable $H_2O_2$ selectivity of >95% and a stable $H_2O_2$ production with a concentration of 0.48 wt% under a high current density of 125 mA cm$^{-2}$ at an applied potential of *ca.* 0.67 V *versus* RHE for 20 h in a flow cell, representing the thus far reported best $H_2O_2$ synthesis COFs electrocatalysts.

Hydrogen peroxide ($H_2O_2$) is an important inorganic chemical and environmentally friendly oxidant with extensive applications in bleaching, disinfection, wastewater treatment, and organic synthesis[1–10]. In industry, anthraquinone method is employed to generate more than 90% of $H_2O_2$, which however is energy-intensive and produces a large amount of toxic by-products[11–13]. For a sustainable future, it is essential to develop an energy efficient and eco-friendly strategy for the synthesis of $H_2O_2$ that should operate onsite even on large or small scales. As a consequence, electrocatalytic 2e$^-$ oxygen reduction reaction (2e$^-$ ORR) has been considered as the most promising alternative approach since it can realize the green and distributed on-demand $H_2O_2$ generation under ambient conditions[14]. Thus far, various electrocatalysts including modified carbon[15–20], noble-metal and alloys[21], non-noble metals[22], metal-organic frameworks (MOFs)[23], and covalent organic frameworks (COFs)[24] have been developed to promote 2e$^-$ ORR for $H_2O_2$ electrosynthesis. However, large-scale electrocatalytic $H_2O_2$ production is still hard to be realized because of the limited solubility of oxygen in electrolyte solutions and easy decomposition of $H_2O_2$ especially in the presence of metal active centers, which usually result in small working currents (<100 mA cm$^{-2}$) and low $H_2O_2$ concentration (<0.1 wt%)[25]. In addition, the favorable thermodynamics to generate water molecules via the 4e$^-$ pathway inevitably reduces the $H_2O_2$ generation capability during ORR[26,27].

[1]Beijing Advanced Innovation Center for Materials Genome Engineering, Beijing Key Laboratory for Science and Application of Functional Molecular and Crystalline Materials, Department of Chemistry and Chemical Engineering, School of Chemistry and Biological Engineering, University of Science and Technology Beijing, Beijing 100083, China. [2]Beijing Synchrotron Radiation Facility, Institute of High Energy Physics, Chinese Academy of Science, Beijing 100049, China. ✉e-mail: kangwang@ustb.edu.cn; liuyunpeng@ihep.ac.cn; jianzhuang@ustb.edu.cn

COFs consist of organic building blocks linked by covalent bonds[28,29], which have drawn great research attention for extensive applications in gas storage and separation[30,31], optoelectronic devices[32], catalysis[33], and energy storage[34] owing to their superiority of high porosity, robust stability, and low density. As a consequence of the ordered pores that favor more exposed active sites to contact with substrate molecules, COFs have shown a great application potentials as promising electrocatalysts for various reactions including ORR[35], oxygen evolution reaction[36,37], hydrogen evolution reaction[38], and $CO_2$ reduction reaction[39]. In particular, two-dimensional (2D) conjugated COFs with ultrastrong fused aromatic linkage have been revealed to exhibit intrinsically high conductivity and thermal/chemical stability, promoting enhanced electrocatalytic performance[40,41]. However, the design of suitable linkers and optimization of reaction conditions for COFs construction remain a demanding task for synthetic chemists. The dioxin[42,43], phenazine[44], and piperazine[45] linkage formation in a 2D conjugated COF by nucleophilic aromatic substitution have been established as fused heterocyclic organic linkage to build up crystalline and stable COFs. Corresponding COFs upon these linkages have been applied for catalysis and energy storage devices. However, these linkages usually result in relatively close layers spacing associated with their typical planar interlayer π-stacking arrangement, leading to the inner active-sites being buried to some degree. Recently, Kaskel[46] and co-workers constructed a dithiine-linked COF with undulated layers due to the bending along the C−S−C bridge but the aromaticity and crystallinity of the overall COF structure still maintained, providing a heuristic for more efficient utilization of buried inner active-sites. In addition to the choice of linkage in 2D conjugated COFs, planar conjugated precursors including porphyrin[47,48], phthalocyanine (Pc)[49], and hexabenzocoronene[50] have usually been selected as building blocks owing to their robust stability and intrinsic high electrical conductivity.

Particularly, metal phthalocyanine (MPc) building units with M-N4 coordination configuration have been demonstrated to act as high-efficiency active sites for catalyzing a series of reactions as exemplified by the efficient 2e− ORR activity of cobalt phthalocyanine (CoPc)[51].

Herein, a dithiine-linked 2D CoPc-based COF, CoPc-S-COF, was afforded from the reaction of hexadecafluorophthalocyaninato cobalt (II) ($CoPcF_{16}$) with 1,2,4,5-benzenetetrathiol (BTT). For the purpose of comparison, a conventional dioxin-linked 2D CoPc-based COF, CoPc-O-COF, was also prepared by reaction between $CoPcF_{16}$ and 1,2,4, 5-tetrahydroxybenzene (THB). Powder X-ray diffraction (PXRD) and electron microscopy analysis results reveal the crystalline porous framework of CoPc-S-COF with an undulated layered structure due to the bending along the C-S-C bridge associated with the large atomic radius and two lone-pairs of electrons of the sulfur atoms in the linking unit, resulting in almost double exposed active Co sites for 2e− ORR compared to CoPc-O-COF with an eclipsed π-stacking model according to the electrochemical analysis. This, in combination with the activated 2e− ORR but deactivated $H_2O_2$ decomposition capability of the same Co center due to the electron-donating effect of S atoms, enables CoPc-S-COF to display a superior electrocatalytic 2e− ORR performance with a remarkable $H_2O_2$ selectivity of >95% and a stable $H_2O_2$ production under a high current density of 125 mA cm−2 at an applied potential of ca. 0.67 V versus RHE for 20 h in a flow cell, generating $H_2O_2$ solution with a concentration of 0.48 wt%.

## Results

### Materials synthesis and characterization

The synthesis of CoPc-O-COF and CoPc-S-COF is illustrated in Fig. 1a and their simulated structural models are displayed in Fig. 1b-e. Nucleophilic substitution reaction of $CoPcF_{16}$ with THB and BTT, respectively, in dimethylacetamide (DMAC) and p-xylene with

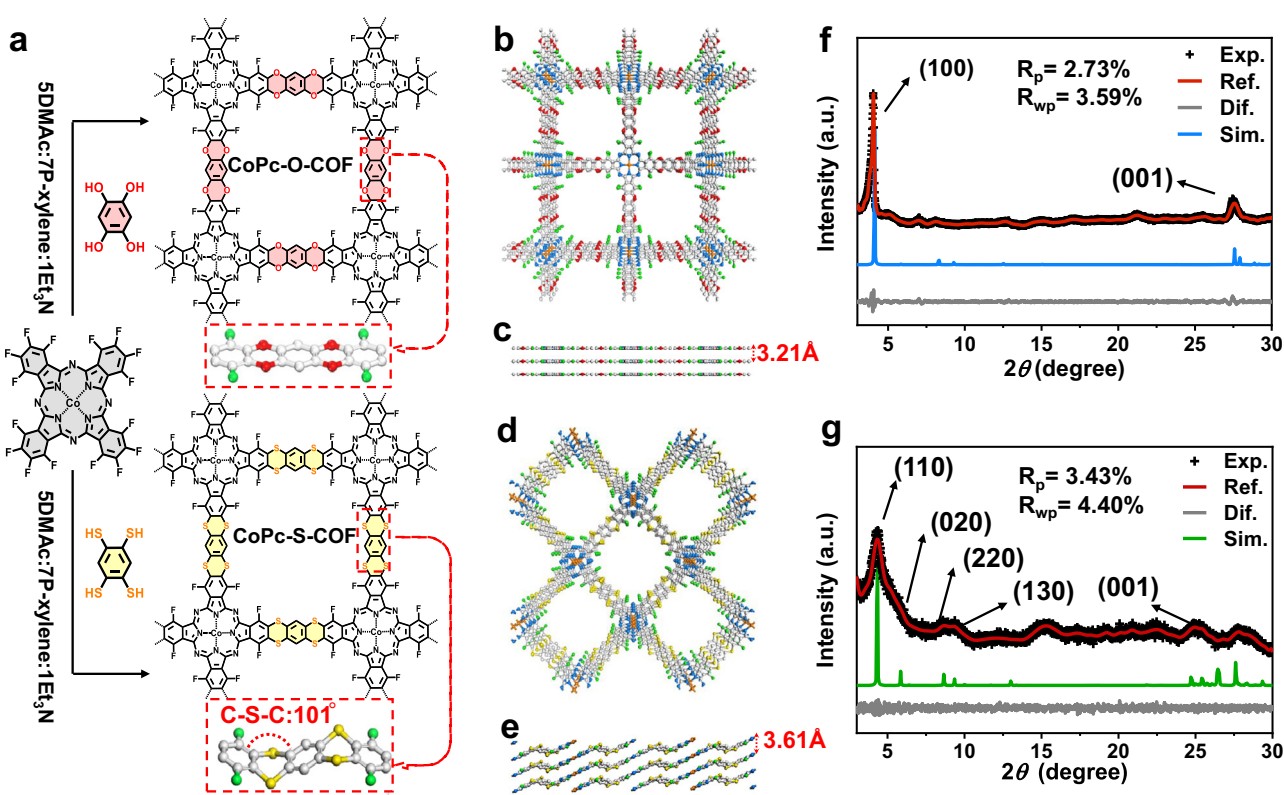

**Fig. 1 | Schematic of synthetic COFs and structural representations of COFs.**
**a** The synthesis route of CoPc-O-COF and CoPc-S-COF. The simulated AA stacking of **b**, **c** CoPc-O-COF and **d**, **e** CoPc-S-COF (Co: orange; C: light gray; N: blue; O: red; S: yellow; F: green). PXRD of **f** CoPc-O-COF and **g** CoPc-S-COF: experimental PXRD profile (black), refined profile (red), the difference between the experimental and refined PXRD (gray), and simulation pattern based on the AA stacking manner (blue and green).

triethylamine (Et₃N) as catalyst affords CoPc-O-COF and CoPc-S-COF in the yield of 75 and 88%. Observation of the band at 1298 cm⁻¹ due to the C-O-C bonds[42] in the Fourier-transform infrared (FT-IR) spectrum demonstrates the successful formation of dioxin bridge in CoPc-O-COF, Supplementary Fig. 1. The characteristic band of the C-S-C linkage[46] gets appeared at 717 cm⁻¹ in the FT-IR spectrum, Supplementary Fig. 2, verifying the successful formation of dithiine bridge in CoPc-S-COF. The solid-state ¹³C cross-polarization/magic-angle-spinning (CP/MAS) NMR spectroscopy reveals the characteristic aromatic carbon signals at 149 and 143 ppm for CoPc-O-COF and CoPc-S-COF, respectively, further supporting the generation of the dioxin/dithiine-linked COFs, Supplementary Figs 3 and 4. Both CoPc-O-COF and CoPc-S-COF exhibit a signal at *ca.* −124 ppm in their solid-state ¹⁹F CP/MAS NMR spectra, indicating their same C-F group nature, Supplementary Fig. 5. The decomposition temperature was revealed to be above 300 °C for both COFs according to thermogravimetric analysis, indicating their great thermal stability, Supplementary Fig. 6. Moreover, the PXRD patterns of both COFs recollected after soaking in different solutions including 1 M KOH, 1 M HCl, pure water, THF, DMF, acetone, ethanol and 3% H₂O₂ for three days remain unchanged, unveiling the good chemical stability of both COFs, Supplementary Figs 7 and 8. In particular, both COFs display very similar FTIR spectra and TEM images before and after the soaking treatment in 3% H₂O₂, further proving the durability of these two COFs in H₂O₂ solution, Supplementary Figs. 9 and 10.

The crystalline structures of these two COFs were estimated by PXRD measurement combined with computational simulation. As displayed in Fig. 1f, CoPc-O-COF shows two strong peaks at 4.13 and 27.48°, corresponding to (100) and (001) facets, respectively. Moreover, the experimental pattern of CoPc-O-COF agrees with the calculated one with AA layer stackings on the basis of Forcite geometrical simulation method, Fig. 1b, c. Furthermore, the Pawley-refined PXRD pattern of CoPc-O-COF using the *P4/mmm* space the observed experimental curve as proved by the good agreement factors of $R_p = 2.73\%$ and $R_{wp} = 3.59\%$, Fig. 1f. The PXRD pattern of CoPc-S-COF exhibits one strong peak at 4.32° and four medium intensity reflections at 6.16, 8.66, 9.46, and 24.73°, corresponding to (110), (020), (220), (130), and (001) facets. Combination of the theoretical simulation and Pawley refinement indicates that CoPc-S-COF adopts undulated layer-stacked structure owing to the nonplanar configuration of the C-S-C units with a dihedral angle of *ca.* 101°, affording the lattice parameters of $a = b = 28.65$ Å, $c = 3.61$ Å, $\alpha = \gamma = 90.00°$, and $\beta = 89.15°$ in *C2/m* space group with $R_p = 3.43\%$ and $R_{wp} = 4.40\%$, Fig. 1c, f, g. The single crystal structure of the model compounds with dithiine moieties provides additional support for the nonplanar configuration of CoPc-S-COF, Supplementary Fig. 11 and Supplementary Table 1. The Raman spectra of both COFs were recorded to explore their vibrational splittings based on corresponding functional bonds, Supplementary Fig. 12. The obvious symmetric peaks due to the aromatic carbon and carbon-oxygen bonds indicate the in-plane vibration nature for CoPc-O-COF. Observation of the series of asymmetric stretching bands due to the aromatic carbon and carbon-sulfur vibrations for CoPc-S-COF indicates the different vibrational splittings of the energy states of the C-S bond associated with the bending or out-of-plane twisting of the bonds[46]. Actually unlike the dioxin-linked structure for CoPc-O-COF, the C-S-C units in CoPc-S-COF are stabilized in a nonplanar configuration to minimize the lone pair electron repulsion of sulfur atoms with large atomic radius in neighboring layers[52], resulting in its undulated layer-stacked structure.

The morphology of CoPc-O-COF and CoPc-S-COF was investigated by scanning electron microscopy (SEM) and transmission electron microscopy (TEM) images, Fig. 2a-f and Supplementary Fig. 13. As can be found, CoPc-O-COF exhibits a micrometer scale irregular sheet morphology, different from the irregular cluster morphology of CoPc-S-COF due to the undulated layer-stacked mode. Both CoPc-O-COF and CoPc-S-COF exhibit distinct lattice fringes with a spacing of $1.93 \pm 0.11$ and $1.88 \pm 0.12$ nm in their high-resolution TEM (HR-TEM) images, which are attributed to the (100) plane of CoPc-O-COF and (110) plane of CoPc-S-COF, respectively, and in turn confirm their high crystallinity, Fig. 2b, e. In addition, clear lattice fringes belonging to the (001) plane of these two COFs get appeared at *ca.* $0.32 \pm 0.02$ nm for CoPc-O-COF and *ca.* $0.37 \pm 0.02$ nm for CoPc-S-COF. Nevertheless, the corresponding fast Fourier transform (FFT) analysis displays the crystalline spot, Supplementary Fig. 14, further demonstrating their good crystallinity. Energy dispersive X-ray (EDX) mapping analysis reveals the elemental composition of Co, C, N, and F in both COFs as well as O element in CoPc-O-COF and S element in CoPc-S-COF with corresponding atom ratios close to the theoretical values, Fig. 2c, f, Supplementary Fig. 15, and Supplementary Table 2. As displayed in Supplementary Figs 16 and 17, N2 adsorption-desorption measurements reveal their permanent porosity with a Brunauer-Emmett-Teller (BET) surface area of 183 m² g⁻¹ for CoPc-O-COF and 285 m² g⁻¹ for CoPc-S-COF with their calculated pore volume of 0.09 cm³ g⁻¹ and 0.14 cm³ g⁻¹, respectively. The pore size distribution of CoPc-O-COF and CoPc-S-COF is determined to be 1.5-2.3 and 1.3-2.1 nm, respectively, with an average pore size of 1.7 and 1.5 nm.

X-ray photoelectron spectroscopy (XPS) was also performed to explore the elemental composition and metal valence states in both COFs. The XPS spectra of both COFs disclose the peaks due to Co, C, N, and F elements, Supplementary Fig. 18 and Supplementary Table 3, in agreement with corresponding EDX mapping results. The Co 2*p* XPS spectrum of CoPc-O-COF displays two peaks at 781.5 and 796.7 eV, attributed to Co 2$p_{3/2}$ and Co 2$p_{1/2}$ of Co (II), Fig. 2g. Nevertheless, the Co 2$p_{3/2}$ and Co 2$p_{1/2}$ peaks of CoPc-S-COF shift to a lower binding energy of 780.8 and 796.1 eV compared to CoPc-O-COF, due to the significant electron-donating effect of S atoms in CoPc-S-COF. The characteristic peaks due to the C-O-C[53] and chemisorbed H₂O/O₂ appear at 531.5 and 533.3 eV, respectively, in the high-resolution O 1s XPS spectrum of CoPc-O-COF, and two peaks centered at 164.8 and 163.5 eV due to C-S-C[54] are observed in the high-resolution S 2*p* XPS spectrum of CoPc-S-COF, Fig. 2h, i, further confirming the successful generation of dioxin/dithiine-connected CoPc-based COFs. Both CoPc-O-COF and CoPc-S-COF exhibit a F 1s peak at 687.2 eV in their F 1*s* XPS spectra, respectively, indicating their same C-F group nature, Supplementary Fig. 19. Additionally, the X-ray absorption near-edge structure (XANES), and extended Xray absorption fine structure (EXAFS) spectra were collected to determine the chemical state and local coordination environment of the Co species. As displayed in Supplementary Fig. 20, the average oxidation state of Co centers in both COFs is close to +2 according to their similar absorption edge to that for CoPc in Co *K*-edge XANES spectra. It is noteworthy that CoPc-S-COF displays the pre-edge peak at 7710 eV, smaller than that of CoPc-O-COF (7711 eV), indicating the oxidization state of Co shifts to a smaller value after S introduction. Morever, the Co L₂, ₃-edge XANES spectra of CoPc-O-COF and CoPc-S-COF were also collected, Supplementary Fig. 21. As shown, the corresponding photon energies of the L₂, ₃ edge white line peaks for CoPc-S-COF are 0.5 eV smaller than those of CoPc-O-COF, consisting with the results of XPS and K-edge XAFS analysis, confirming the oxidization state of Co shifts to a smaller value after S introduction. These results further suggest the S atoms are able to modulate the electron structure of the atomic Co active sites, in line with the XPS results. Moreover, the Fourier transform (FT) EXAFS spectra and corresponding fitting results of both COFs and CoPc show peaks at *ca.* 1.53, 2.49, and 3.10 Å are attributed to the Co-N (first shell), Co-C (second shell), and Co-N (third shell) scattering paths with a coordination number of 4, 8, and 4, Supplementary Figs 22, 23 and Supplementary Table 4. In particular, the Co content of CoPc-O-COF and CoPc-S-COF amounts to 5.91 and 5.01 wt%, respectively, according to inductively coupled plasma-optical emission spectrometry (ICP-OES), very close to the theoretical values, Supplementary Table 5.

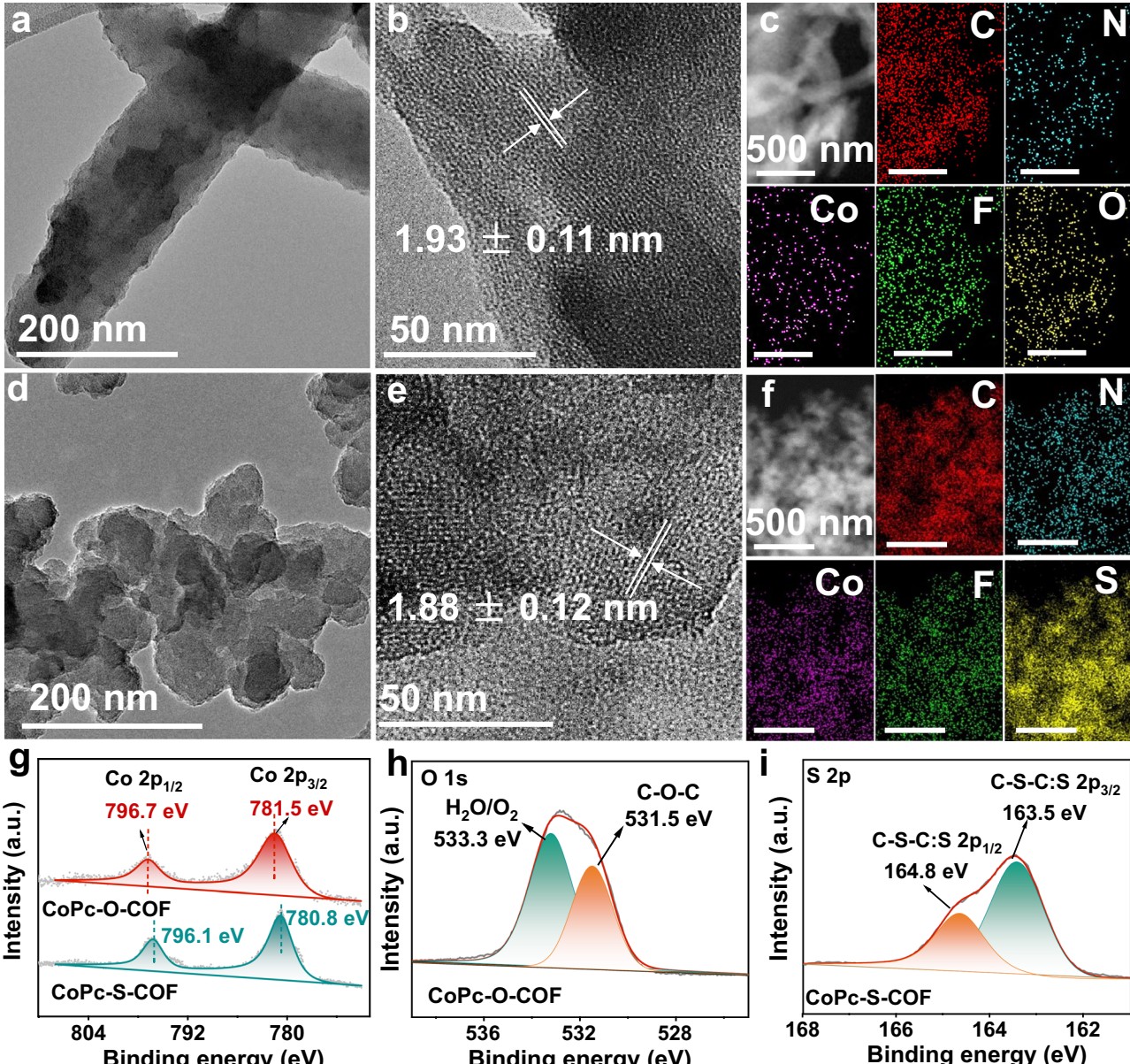

**Fig. 2 | Morphology and characterization of COFs.** TEM images of **a** CoPc-O-COF and **d** CoPc-S-COF. HR-TEM images of **b** CoPc-O-COF and **e** CoPc-S-COF. The EDX mapping analysis of **c** CoPc-O-COF and **f** CoPc-S-COF. High-resolution XPS spectra of **g** Co 2*p* for CoPc-O-COF and CoPc-S-COF, **h** O 1*s* for CoPc-O-COF and **i** S 2*p* for CoPc-S-COF.

## Electrocatalytic ORR performance

The ORR measurements were performed on the three-electrode system with the rotating ring-disk electrode (RRDE) used as the working electrode in the condition of alkaline media. As shown in Fig. 3a, the ORR polarization curves and $H_2O_2$ detection current of both COFs are collected on RRDE at 1600 rpm in $O_2$-saturated 0.1 M KOH. CoPc-S-COF electrodes offer high electrocatalytic activity for oxygen reduction to render an onset potential of 0.81 V *versus* reversible hydrogen electrode (RHE) (defined at −0.1 mA cm$^{-2}$ of $H_2O_2$ partial current)[26,55], higher than that of CoPc-O-COF electrode (0.78 V). Moreover, the Tafel slopes of CoPc-O-COF and CoPc-S-COF are calculated to be 62 and 55 mV dec$^{-1}$, respectively, smaller than that of individual CoPcF$_{16}$ (75 mV dec$^{-1}$), Supplementary Fig. 24. This indicates the faster 2e$^-$ ORR kinetics of both COFs, which might be attributed to their high initial electron transfer efficiency and large active surface during the catalytic process. Figure 3b presents the $H_2O_2$ selectivity and electron transfer number *n* during ORR for both COFs. The $H_2O_2$ selectivity of CoPc-S-

COF amounts to larger than 90% in the potential range of 0.20–0.70 V *versus* RHE with an *n* value of 2.0–2.2, suggesting its promising 2e$^-$ ORR performance. Meanwhile, the $H_2O_2$ selectivity value of CoPc-O-COF is slightly lower than that of CoPc-S-COF in the same potential range. According to the ORR polarization curves at different rotation rates and Koutecky–Levich (K–L) equation, the electron transfer number *n* of CoPc-O-COF and CoPc-S-COF is determined to be *ca.* 2.3 and 2.0, respectively, Supplementary Fig. 25, consistent with the RRDE result. The Co mass activity (MA) of both COFs was also calculated, Supplementary Fig. 26. CoPc-S-COF exhibits a MA of 80 A g$_{Co}^{-1}$ at 0.7 V vs RHE, superior to that of CoPc-O-COF, 48 A g$_{Co}^{-1}$ at 0.7 V vs RHE. In addition, the conductivity of both COF electrodes was analyzed by electrochemical impedance spectroscopy (EIS) measurements. As shown in Supplementary Fig. 27, CoPc-O-COF and CoPc-S-COF electrodes exhibit a small EIS semicircle diameter of 112 and 87 Ω, indicating the optimized charge transfer of these two COFs owing to their conjugated structure. Nevertheless, the double-layer capacitances ($C_{dl}$) of both

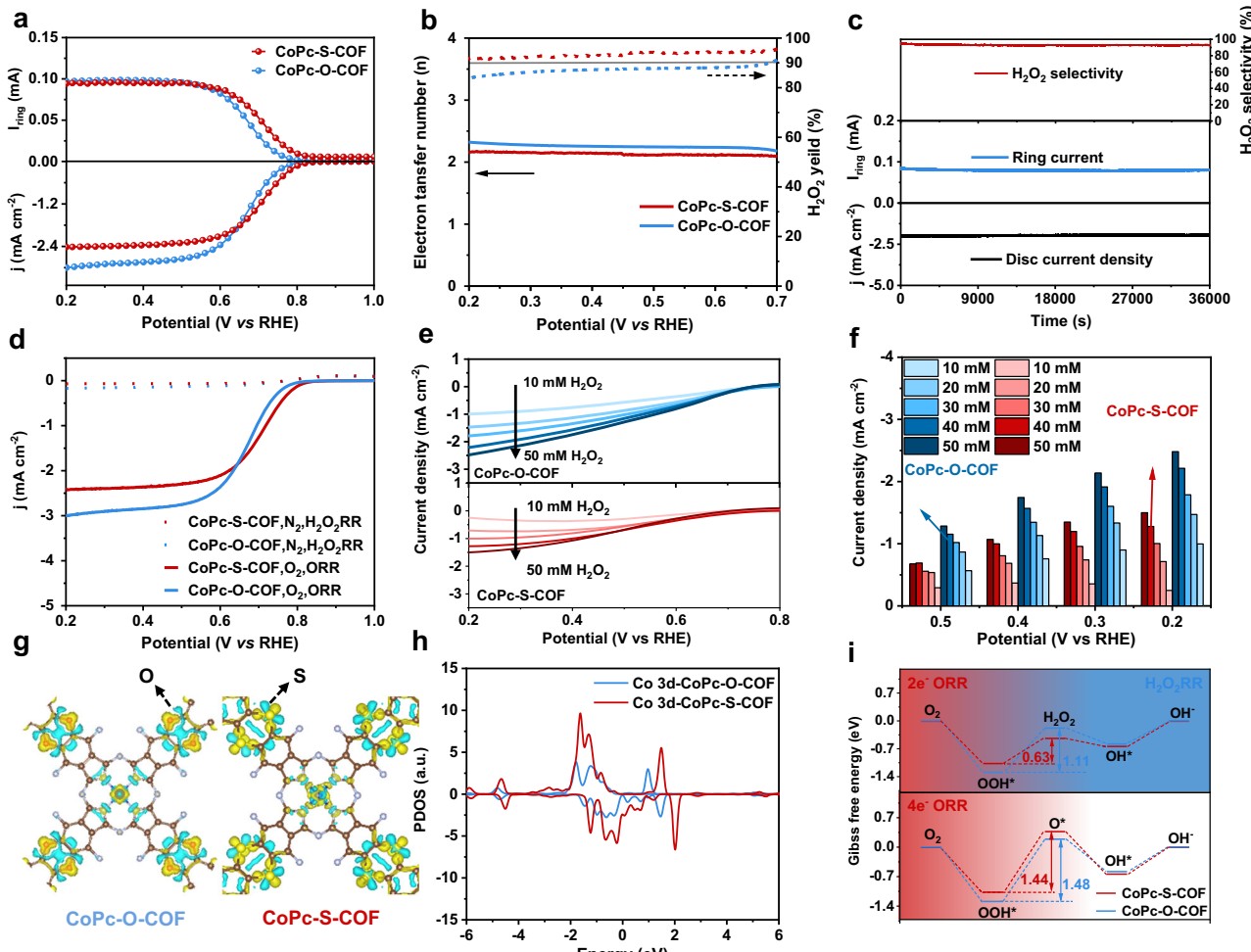

**Fig. 3 | $H_2O_2$ electroproduction and DFT calculation. a** LSVs of CoPc-O-COF and CoPc-S-COF at 1600 rpm in $O_2$-saturated 0.1 M KOH. **b** $H_2O_2$ selectivity and electron transfer number $n$ of CoPc-O-COF and CoPc-S-COF. **c** Chronoamperometry measurement of CoPc-S-COF for 36000 s at 0.52 V versus RHE. **d** ORR polarization curves in $O_2$-saturated 0.1 M KOH and $H_2O_2RR$ polarization curves in Ar-saturated 0.1 M KOH containing 1 mM $H_2O_2$. **e** $H_2O_2RR$ polarization curves and **f** current densities of CoPc-O-COF and CoPc-S-COF in Ar-saturated 0.1 M KOH containing different concentrations of $H_2O_2$ (Catalyst mass loading: 0.16 mg cm$^{-2}$, 0.1 M KOH: pH = 13). **g** Differential charge distribution on both simulated periodic fragment of both COFs (isosurface Level=0.01). **h** Partial density of states (PDOS) of Co 3d-orbital in different models. **i** Reaction free energy change for 2e$^-$ ORR, 4e$^-$ ORR and $H_2O_2RR$ process of both COFs.

COFs, which is proportional to their electrochemical surface area, were derived from the CV curves at different sweep rates, Supplementary Fig. 28. The $C_{dl}$ of CoPc-S-COF is calculated as 125 μF cm$^{-2}$, much larger than that of CoPc-O-COF, 73 μF cm$^{-2}$, indicating the more available active sites within CoPc-S-COF originated from its twisted layered structure, in turn leading to its higher 2e$^-$ ORR performance. In line with this point, the surface electrochemical active sites on the CoPc-S-COF electrode are calculated to be 62.3 nmol cm$^{-2}$ according to the peak current of CV curves as a function of scan rate[56], Supplementary Fig. 29, revealing 11.0% of the total cobalt-phthalocyanine units acting as active sites. This value is almost twice of that for the CoPc-O-COF electrode, 6.9%, confirming the more exposed active sites in CoPc-S-COF, Supplementary Fig. 30. This in turn becomes responsible for the much superior 2e$^-$ ORR activity of CoPc-S-COF to CoPc-O-COF.

The catalytic stability of both COFs on RRDE was assessed by the chronoamperometry measurements, Fig. 3c and Supplementary Fig. 31. The Co-S-COF electrode shows almost unchanged current signals and maintained $H_2O_2$ selectivity of >93% for 10 h at 0.52 V versus RHE, verifying its robust catalytic durability for 2e$^-$ ORR in alkaline media. As displayed in Supplementary Fig. 31, Co-O-COF also shows good catalytic stability on RRDE. Moreover, both CoPc-O-COF and CoPc-S-COF exhibit similar PXRD patterns before and after the

stability tests, proving their robust framework nature, Supplementary Fig. 32. The scanning transmission election microscope and XANES measurements were also carried out for both COFs before and after the stability tests. As exhibited in Supplementary Fig. 33, no Co nanoparticles can be observed in the STEM images of the two COFs before and after the stability tests, excluding the Co leaching and aggregation during the electrochemical test. Particularly, the Co K-edge EXAFS spectra of both COFs after the stability tests also show only the peak at 1.53 Å due to the Co-N scattering path without giving the peak at 2.20 Å due to the Co-Co bond, further confirming the absence of Co nanoparticles formed from the Co leaching during the electrochemical test and in turn the high durability of the two COFs, Supplementary Fig. 34. The poison experiment was then carried out to identify the catalytic site by using SCN$^-$ ions, which tend to bind the metal atoms of CoPc and block the adsorption of reaction intermediates, thus negatively affecting the catalytic performance. As shown in Supplementary Fig. 35, significant degradation of the catalytic activity occurs after adding SCN$^-$ into the electrolyte, demonstrating the actual active site nature of the Co metal center for 2e$^-$ ORR in the two COFs[57]. Additionally, in-situ Co center poisoning experiments with nitrite has also been carried out and shown in Supplementary Fig. 36. As can be found, significant degradation of the

catalytic activity for CoPc-S-COF occurs after adding nitrite into the electrolyte, further confirming the active site nature of the Co atoms towards 2e⁻ ORR in CoPc-S-COF. Particularly, the polarization curves of $H_2O_2$ reduction reaction ($H_2O_2RR$) of both COFs were recorded in the Ar-saturated 0.1 M KOH solution containing 1 mM $H_2O_2$. As revealed in Fig. 3d, both samples display negligible $H_2O_2RR$ activity. To more clearly compare the inhibition of $H_2O_2$ decomposition for both COFs, the $H_2O_2RR$ polarization curves of CoPc-O-COF and CoPc-S-COF in Ar-saturated 0.1 M KOH solutions containing different concentrations of $H_2O_2$ were measured, Fig. 3e, f. CoPc-O-COF exhibits larger $H_2O_2RR$ currents than those of CoPc-S-COF especially in high $H_2O_2$ concentration (10 mM-50 mM). This result indicates the superior inhibition of CoPc-S-COF to $H_2O_2$ decomposition during the 2e⁻ ORR process due to the enriched electron nature of the Co center associated with the electron-donating sulfur atoms in the linking unit, favouring the production of high concentration $H_2O_2$ solution.

To explore the impact of F atom on the catalytic activity and selectivity of the Co centers in CoPc-O-COF and CoPc-S-COF, the catalytic activity of the small molecule Co phthalocyanines including CoPc, $CoPcF_{16}$, and the metal-free phthalocyanine $H_2PcF_{16}$ electrodes have also been prepared and tested on RRDE at 1600 rpm in $O_2$-saturated 0.1 M KOH, Supplementary Fig. 37. It is worth noting that the CoPc and $CoPcF_{16}$ electrodes exhibit the same onset potential of 0.78 V *versus* RHE and similar $H_2O_2$ selectivity (>60%) in the potential range of 0.20−0.60 V *versus* RHE with an *n* value of 2.5-2.7, demonstrating their obvious similar reactivity for 2e⁻ oxygen reduction with negligible effect of F atoms. However, the metal-free $H_2PcF_{16}$ electrode displays much inferior electrocatalytic performance with lower onset potential of 0.68 V *versus* RHE and $H_2O_2$ selectivity (<50%), confirming the nature of Co active centers. For the purpose of further clarifying the impact of O and S atoms on the catalytic activity and selectivity of the Co centers in CoPc-O-COF and CoPc-S-COF, two small molecule Co phthalocyanines including CoPc-S and CoPc-O containing C-S-C and C-O-C groups, respectively, were synthesized with their 2e⁻ ORR performance assessed, Supplementary Figs 38 and 39. As can be seen, CoPc-S shows better 2e⁻ ORR activity and worse $H_2O_2RR$ activity in comparison with CoPc-O, proving the effect of S on enhancing the 2e⁻ ORR performance of CoPc-S-COF. In addition, the S-doped CoPc-O-COF (named CoPc-O-COF-S 20%) was prepared with its electrocatalytic performance tested. As expected, CoPc-O-COF-S 20% shows high ORR activity with an onset potential of 0.80 V versus RHE and an electron transfer number of 2.2-2.3, Supplementary Figs. 40–42. Nevertheless, CoPc-O-COF-S 20% displays smaller $H_2O_2RR$ currents than those of CoPc-O-COF in the Ar-saturated 0.1 M KOH solution containing 10-40 mM $H_2O_2$, revealing the effect of S-doping on diminishing the catalytic activity towards $H_2O_2$ decomposition.

To gain further insight into the 2e⁻ ORR activity of the COFs, the electronic structure of CoPc-O-COF and CoPc-S-COF was investigated by density functional theory (DFT) calculation. The adsorption energy ($\Delta G_{ads}$) of $O_2$ on various atoms of the two COFs including Co, N, O, S, and F has been firstly calculated to explore the active site of $H_2O_2$ production. As can be seen in Supplementary Fig. 43 and Supplementary Table 6, the significantly smaller value of $\Delta G_{ads}$ for Co atom, *ca.* −0.3 eV, than those for other atoms (*ca.* −0.02 to −0.05 eV for N, *ca.* +0.4-+0.6 eV for O/S/F) reveals the active site nature of Co atom in the two COFs towards ORR. In addition, the electron density on the CoPc moiety including around Co site in CoPc-S-COF is higher than that for CoPc-O-COF due to the electron-donating effect of S atoms, which facilitates the charge transfer between Co active sites and intermediates and in turn affords enhanced catalytic activity for 2e⁻ ORR[58], Fig. 3g and Supplementary Fig. 44. Moreover, the natural population analysis (NPA)[59] charge distribution calculation of both COFs was also carried out to assess the electron transferring process in both COFs. As displayed in Supplementary Table 7, the change of atomic charges on the conjugated chain is in a wave manner of S ↑ -$C_a$ ↓ -$C_b$ ↑ -$C_c$ ↓ -$C_d$ ↑ -

(NCo)↓ during the electron transferring process from S/O atoms to central Co atoms, exhibiting an alternative polarization effect along the π electron transferring pathway in both COFs[59]. In particular, the central Co atom gains an additional charge of -30 × |$10^{-3}e$| when the O atoms in CoPc-O-COF are replaced by S atoms in CoPc-S-COF, further confirming a more obvious charge transfer from S to Co compared to that from O to Co. Furthermore, the calculated projected density of states (pDOS) discloses a lower d band center position of −1.33 eV for CoPc-S-COF with higher intensity of peaks near the Fermi level ($E_f$) compared to CoPc-O-COF with a d band center position of −1.17 eV, Fig. 3h. This indicates that larger density of active electrons around Co centers in CoPc-S-COF participates in the electrochemical ORR reaction, confirming the higher catalytic activity of CoPc-S-COF due to the electron-donating effect of S atoms. Nevertheless, Fig. 3i presents the calculated Gibbs free energy differences (ΔG) diagrams of the 2e⁻ ORR and 4e⁻ ORR processes on CoPc-S-COF and CoPc-O-COF. As can be found, the conversion of OOH* to $H_2O_2$ is the rate-determining step of 2e⁻ ORR on both CoPc-S-COF and CoPc-O-COF with an energy barrier of 0.63 and 1.11 eV, respectively. These values are smaller than the energy barrier of the OOH* to O* conversion process (the rate-determining step of 4e⁻ ORR) on both COFs, 1.44 and 1.48 eV, demonstrating the faster reaction kinetics of 2e⁻ ORR than 4e⁻ ORR on both COFs and in turn their high selectivity towards 2e⁻ ORR. In particular, the lower OOH* to $H_2O_2$ conversion energy barrier of CoPc-S-COF in comparison with that for CoPc-O-COF indicates the enhanced 2e⁻ ORR activity of the former species over the latter one, while the higher $H_2O_2$ to OH⁻ conversion energy barrier of CoPc-S-COF than that for CoPc-O-COF illustrates the diminished activity towards $H_2O_2$ decomposition of CoPc-S-COF to CoPc-O-COF, rationalizing the more stable electrocatalytic $H_2O_2$ production of CoPc-S-COF.

Gas diffusion electrode (GDE) devices were used to further explore the practical application potential of the as-prepared CoPc-based COFs towards 2e⁻ ORR. A three-phase flow cell, Fig. 4a, in which the catalyst is deposited on a gas diffusion layer (GDL) as the work electrode, is deemed to be able to afford higher reduction current densities by increasing oxygen concentration on GDL and improve the $H_2O_2$ production rate. Corresponding measurements of both COFs were carried out in 1 M KOH. As exhibited in Fig. 4b, both COFs show high electrocatalystic activity with much higher current density in flow cell compared to RRDE measurements. The chronoamperometry measurements at varied applied voltages were conducted with the generated $H_2O_2$ determined by the $Ce^{4+}$ titration method[60], Fig. 4c, and Supplementary Figs. 45 and 46. Figure 4d shows the determined Faradaic efficiency of $H_2O_2$ ($FE_{H2O2}$) for both COFs. CoPc-O-COF and CoPc-S-COF could exhibit over 80% $FE_{H2O2}$ in the range of 0.73 to 0.33 V *versus* RHE, even higher than the RRDE measurements in the range from 0.73 to 0.53 V. Remarkably, the $J_{H2O2}$ of CoPc-S-COF reaches 152 mA cm⁻² at 0.63 V *versus* RHE (equivalent to an overpotential of 130 mV) with a $FE_{H2O2}$ of 98 % in 1 M KOH at room temperature, which gets further increased to 415 mA cm⁻² at 0.33 V *versus* RHE (equivalent to an overpotential of 430 mV) with $FE_{H2O2}$ still higher than 80%, superior to CoPc-O-COF, Fig. 4c, d, and Supplementary Fig. 45. The $H_2O_2RR$ performance of both COFs was also explored in the flow cell. As displayed in Fig. 4b, CoPc-S-COF shows much smaller $H_2O_2$ reduction current densities compared to CoPc-O-COF, consistent with the result in the RRDE system. This actually suggests that $H_2O_2RR$ could be inhibited in CoPc-S-COF and therefore leads to higher $FE_{H2O2}$ and $H_2O_2$ concentration. Furthermore, the long standing and stable $H_2O_2$ production of CoPc-S-COF has been recorded at a fixed current density of 125 mA cm⁻², Fig. 4e, f. CoPc-S-COF maintains $FE_{H2O2}$ > 95% in the continuous $H_2O_2$ electroproduction for 20 h. Nevertheless, the $H_2O_2$ amount produced gets linearly increased along with increasing the operating time with an almost constant production rate of *ca.* 9500 mmol $g_{cat}^{-1}$ h⁻¹ and an almost unchanged operating voltage of *ca.* 0.67 V *versus* RHE, Fig. 4f, further confirming the high stability of

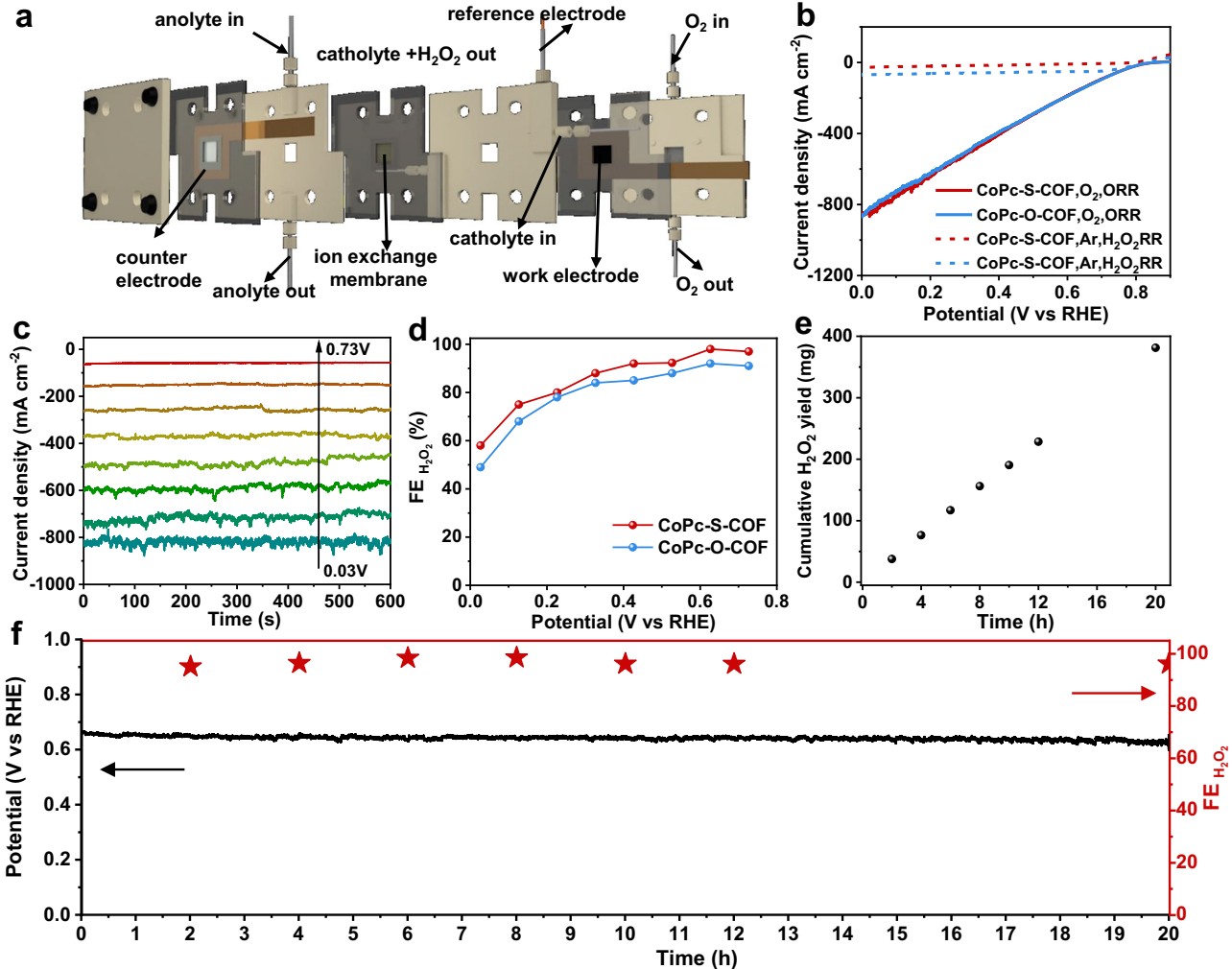

**Fig. 4 | H₂O₂ electroproduction in the flow cell. a** Schematic diagram of the flow cell. **b** LSVs of CoPc-O-COF and CoPc-S-COF in flow cell. **c** The chronoamperometry measurements at varied applied voltages of CoPc-S-COF. **d** FE$_{H2O2}$ of CoPc-O-COF and CoPc-S-COF at varied applied voltages. **e** H₂O₂ yields of CoPc-S-COF.

**f** Chronopotentiometry curve at a current density of 125 mA cm⁻² and the corresponding FE$_{H2O2}$ in the flow cell for CoPc-S-COF (Catalyst mass loading: 0.48 mg cm⁻², 1 M KOH: pH = 14).

CoPc-S-COF as well as its inactivation towards H₂O₂ decomposition. More importantly, the H₂O₂ yield reaches up to 377 mg after 20 h electrolysis corresponding to a H₂O₂ concentration of 0.48 wt%, Fig. 4e. It is worth noting that the features of COFs such as the molecularly structural tunability and desired permanent porosity enable this class of materials to be promising advanced catalysts with well-defined structure and active sites for various electrochemical applications[61]. Recently, 2D COFs such as PYTA-TPEDH-COF[62], TP-TD-COF[63], and Py-TD-COF[64] as well as 3D COFs including BUCT-COF-1[65] and BUCT-COF-7[66] have been employed as metal-free electrocatalysts to promote 2e⁻ ORR with H₂O₂ selectivity up to *ca*. 90%. Introduction of highly efficient Co active sites for 2e⁻ ORR into CoPc-S-COF affords its enhanced H₂O₂ selectivity (>95%) in the present case. Nevertheless, the undulated layer-stacked structure of CoPc-S-COF favours the exposure of active sites, resulting in the higher H₂O₂ production rate in comparison with all the thus far reported COF-based electrocatalysts, Supplementary Table 8. Actually, the electrocatalytic H₂O₂ production performance of CoPc-S-COF is also competitive with that of the state-of-the-art inorganic catalytic materials, Supplementary Table 8, demonstrating its great application potential towards practical H₂O₂ production. Moreover, after the stability test, CoPc-S-COF shows constant structure according to the FTIR and XPS analysis, further proving its robust stability, Supplementary Figs. 47 and 48.

To further reveal the application potential of CoPc-S-COF, the electrocatalytic H₂O₂ production activity of CoPc-S-COF has also been evaluated in 1 M Na₂SO₄ (pH = 7) in the GDE device. As shown in Supplementary Fig. 49, CoPc-S-COF displays high 2e⁻ ORR activity in the neutral electrolyte with a high H₂O₂ selectivity (*ca*. 80%) in the potential range of 0.03–0.33 V versus RHE. More importantly, CoPc-S-COF displays a stable H₂O₂ production with a high FE$_{H2O2}$ of *ca*. 80% under a current density of 100 mA cm⁻² at an applied potential of *ca*. 0.30 V vs RHE for 20 h, generating 256 mg H₂O₂, comparable to the result obtained in alkaline electrolyte. These additional experimental results further confirm the good electrocatalytic H₂O₂ production performance of CoPc-S-COF, which is beneficial to the design and preparation of high-performance and low-cost electrocatalysts towards industrial-level H₂O₂ electroproduction.

## Discussion

In summary, a porous dithiine-linked CoPc-based COF was fabricated. CoPc-S-COF possesses an undulated layer-stacked structure due to the bending along the C-S-C bridge to allow more exposed Co centers for 2e⁻ ORR. This, in combination with the activated 2e⁻ ORR but deactivated H₂O₂ decomposition capability of the Co center because of the electron-donating effect of S atoms, enables CoPc-S-COF to display a high H₂O₂ selectivity and realize the large-scale H₂O₂ production in the

flow cell. This work should be beneficial to the design and preparation of high-performance and low-cost electrocatalysts for $H_2O_2$ electrosynthesis.

## Methods

### Synthesis of CoPc-O-COF

$CoPcF_{16}$ (8.6 mg, 0.01 mmol) and THB (2.8 mg, 0.02 mmol) were added into the mixed solvent of 0.7 mL p-xylene and 0.5 mL DMAc in a 16 mL Pyrex tube. The mixture was sonicated for 5 min to form a homogeneous suspension. Then 100 µL triethylamine was added into the mixture. After three freeze-pump-thaw cycles, the Pyrex tube was sealed and heated in an oven at 100 °C for 7 days. The black-green precipitate was collected by centrifugation and rinsed with acetone, dichloromethane, and THF in a Soxhlet extractor for one day. Finally, CoPc-O-COF was then obtained as black powder in a yield of 75%.

### Synthesis of CoPc-S-COF

$CoPcF_{16}$ (8.6 mg, 0.01 mmol) and BTT (4.2 mg, 0.02 mmol) were added into the mixed solvent of 0.7 mL p-xylene and 0.5 mL DMAc in a 16 mL Pyrex tube. The mixture was sonicated for 5 min to form a homogeneous suspension. Then 100 µL triethylamine was added into the mixture. After three freeze-pump-thaw cycles, the Pyrex tube was sealed and heated in an oven at 100 °C for 7 days. A dark-green solid was formed inside the Pyrex tube during the reaction process. The product was collected by centrifugation and then solvent exchanged with dry ethanol for a week followed by supercritical drying. Finally, CoPc-S-COF was obtained as fluffy dark-green product in a yield of 88%.

## Data availability

All relevant data that support the findings of this study are presented in the manuscript and supplementary information file. Source data are provided with this paper.

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

## Acknowledgements

Financial support from the Natural Science Foundation of China, grant Nos. 22235001 (J.J.), 22175020 (J.J.) and 12305372 (Y.L.). University of Science and Technology Beijing is gratefully acknowledged. The authors also wish to thank the facility support of the 4B9A beamline of Beijing Synchrotron Radiation Facility (BSRF).

## Author contributions

Conceptualization: Q.Z., K.W., and J.J. Methodology: Q.Z., R.J., and K.W. Investigation: Q.Z., R.J., and Y.L. Visualization: Q.Z. and X.Y. DFT calculation: Y.J. and D.Q. Writing—original draft: Q.Z. Writing—review and editing: Q.Z., R.J., X.Y., Y.J., D.Q., K.W., and J.J.

## Competing interests

The authors declare no competing interests.
