## [Peer review file · Nature Communications]

REVIEWER COMMENTS

Reviewer #1 (Remarks to the Author):

The manuscript presents a research work on the catalytic performance of a Co-Pc based COF catalyst for electrochemical H₂O₂ production. The work demonstrates that Co-Pc based COF with C-S-C linkage has improved selectivity for H₂O₂ production comparing to the one with C-O-C linkage. The DFT calculations attribute this enhancement to the electron-donating nature of S. However, the achieved selectivity and current density is not exceeding published works using similar catalysts (Nat. Commun. (2023)14:172, Nat. Commun.(2023)14:1426, Energy Environ. Sci., 2023,16, 446-459). Besides, the explanation for the improvement seems inconsistent. DFT calculations found Co is more electron-rich with C-S-C linkage. However, OOH adsorption is calculated to be weakened on this electron-rich Co site, which is to my best knowledge not quite making sense because electron-rich site should promote electron transfer to OOH and thus enhances the adsorption. Overall, the work is interesting in modifying COF linkage for tuning its catalytic performance, however, CoPc based H₂O₂ catalysts are not novel and the understanding for the improved performance is not strong enough. So, I would not recommend this work to be published in Nat. Commn.

Reviewer #2 (Remarks to the Author):

In this manuscript, the authors reported a bridge bonded metal phthalocyanine framework with undulated layers for H₂O₂ electrosynthesis. This work is interesting in crosslinking CoPc and COF, but the current manuscript cannot be accepted unless the following issues have been soundly addressed.

1. The supplementary Fig. 29 didn't match the description in main text.
2. In Figure 4c, the current density of the ordinate unit was marked as A cm⁻², which was described as mA cm⁻² in discussion part. The same mistake could also be observed in supplementary Fig. 39.
3. As is known to all, the peak intensity of R space curves directly associates with the coordination number of Co. In supplementary Fig. 17, the peak intensity of CoPc with Co-N₄ configuration is significantly higher than that of CoPc-O-COF and CoPc-S-COF, which demonstrates that the coordination numbers of Co-N for CoPc-O-COF and CoPc-S-COF are lower than that of CoPc. Besides, the EXAFS data should be fitted based on standard sample (e.g. CoPc in this work), but the authors only displays the EXAFS fitting results of CoPc-O-COF and CoPc-S-COF. In addition, the valence states obtained from XANES spectra should compared by standard samples. In this work, charge transfer was obviously observed from the XPS results (Fig. 2g), which may also lead to a significant change in XANES spectra. Hence, the data processing of the XAFS in this manuscript is unprofessional and insufficient to fully reveal the local structure information of the material.

4. The Ag/AgCl reference electrode is not suitable for alkaline conditions, Hg/HgO reference electrode is suggested.

5. In line 234 to 236, the author declares that the higher double layer capacitances of CoPc-O-COF and CoPc-S-COF in turn leading to its higher 2e⁻ ORR performance. How about the relationship between double layer capacitances and ORR selectivity?

6. As illustrated in Fig 3g, the oxygen and sulfur atoms are far away from the cobalt atom, and this electron contribution effect may be diluted by the surrounding carbon and nitrogen atoms, thereby resulting in an insufficiently defined electronic structure for cobalt.

Reviewer #3 (Remarks to the Author):

Manuscript: NCOMMS-23-34167

"Dithiine-linked Metalphthalocyanine Framework with Undulated Layers for Highly Efficient and Stable H₂O₂ Electroproduction", by Qianjun Zhi, Rong Jiang, Xiya Yang, Yucheng Jin, Dongdong Qi, Kang Wang, and Jianzhuang Jiang

In this work, cobalt phthalocyanine (CoPc)-based covalent organic frameworks (COF) are prepared and used for the two-electron oxygen reduction reaction (ORR) to generate hydrogen peroxide. This is a thorough study in which various physico-chemical characterization methods are employed. The results of the electrochemical investigations are reliable. There are still some issues that need to be addressed.

Overall. In current form the paper is not suitable for publication in this highly ranked Journal.

Recommendation: major revision

Comments to the authors:

1. Page 3, lines 41-43. "As a consequence..." This sentence needs a reference to be made (see, for example, Lobyntseva et al., doi:10.1016/j.electacta.2007.05.076). This paper should be cited.

2. Page 3, line 49. This Journal has a wide readership. It would be of particular interest for the readers of this Journal to provide the main types of electrocatalysts used for hydrogen peroxide electrosynthesis. Literature should be consulted in this regard.

3. Page 7, line 164. What could be the reason for a large difference between the BET values of CoPc-O-COF and CoPC-S-COF materials? Pore size distribution should be added to Supplementary Fig. 13.

4. Page 8, line 202. At which current density threshold the ORR onset potential was determined?

5. Page 8, last sentence. The Tafel analysis is also used to discuss about the ORR mechanism on the catalyst materials. The discussion about the mechanism of the ORR should be added.
6. Page 10, lines 225-227. The mass activity (MA) for ORR might depend on the catalyst loading. As usual, the MA values are calculated in the kinetic region of the ORR process. It is pointless to calculate MA at 0.3 V in which the diffusion-limited current plateau is formed and the ORR process is controlled by mass transfer. This review suggests the authors to determine MA at 0.8 V. Therefore, the potential range in Supplementary Fig. 20 should be presented up to 0.75 V.
7. Page 10, line 239. Why is the active amount of CoPc units so low (only 11.0%)?
8. Page 11, lines 259-264 (Supplementary Fig. 29). The poisoning tests were performed in the presence of 1 mM SCN⁻. One can assume that Co-N4 sites are blocked by adding SCN⁻ anions. However, still a substantial ORR current is observed in the presence of 1 mM SCN⁻. What could be those active sites, which catalyze the ORR in 0.1 M KOH containing 1 mM SCN⁻?
9. Page 12, lines 291-293 (Supplementary Fig. 32). What could be the reason for a different ORR activity between CoPc-O and CoPc-S? In Figure 1a CoPc-O-COF and CoPc-S-COF catalysts show rather similar ORR electrocatalytic activity.
10. Page 14, lines 360-362. The authors should perform a thorough physical characterization (post-mortem analysis) of the catalyst materials after soaking them in a peroxide solution for longer time to check the durability of these catalysts in a higher concentration of H₂O₂ solution.

Minor remarks:

Page 3, line 44. 'electrocatalytic' should be 'electrocatalytic' (see also page 12, line 296)

Page 4, line 97. 'trimethylamine (Et₃N)' This is not correct.

Page 8, line 176. "shift to a lower energy" should be "shift to a lower binding energy"

Page 8, line 202. 'high reactivity' should be replaced with 'high electrocatalytic activity'

Page 10, line 223. "(K-L) diffusion equation" should be replaced with "(K-L) equation"

Page 14, line 346. "excellent catalytic activity" should be "excellent electrocatalytic activity"

Supplementary material, page 4, line 79. KV should be kV as the unit of voltage

Supplementary material, page 6, line 130. "to respond to" should be "to detect"

Supplementary material, page 9, line 213. 'chronopotentiometry' should be 'chronoamperometry'

Response to reviewers' reports:

Reviewer #1 (Remarks to the Author):

The manuscript presents a research work on the catalytic performance of a Co-Pc based COF catalyst for electrochemical H₂O₂ production. The work demonstrates that Co-Pc based COF with C-S-C linkage has improved selectivity for H₂O₂ production comparing to the one with C-O-C linkage. The DFT calculations attribute this enhancement to the electron-donating nature of S. However, the achieved selectivity and current density is not exceeding published works using similar catalysts (Nat. Commun. (2023)14:172, Nat. Commun.(2023)14:1426, Energy Environ. Sci., 2023,16, 446-459). Besides, the explanation for the improvement seems inconsistent. DFT calculations found Co is more electron-rich with C-S-C linkage. However, OOH adsorption is calculated to be weakened on this electron-rich Co site, which is to my best knowledge not quite making sense because electron-rich site should promote electron transfer to OOH and thus enhances the adsorption. Overall, the work is interesting in modifying COF linkage for tuning its catalytic performance, however, CoPc based H₂O₂ catalysts are not novel and the understanding for the improved performance is not strong enough. So, I would not recommend this work to be published in Nat. Commn.

Answer: First of all, we truly appreciate this reviewer's valuable comments towards improving the quality of the present research. The electrocatalytic oxygen reduction reaction (ORR) to H₂O₂ via a two electron (2e⁻) pathway offers an economical, safe, and environmentally friendly route to H₂O₂. However, the actual electrochemical H₂O₂ syntheses process is challenging due to the fact that many electrode materials favor the competitive four-electron (4e⁻) ORR to H₂O, rather than the 2e⁻ ORR to H₂O₂, thus reducing the yield of H₂O₂. It is therefore of great importance to discover more ORR electrocatalysts with high selectivity to H₂O₂. The references mentioned by this reviewer indeed have reported similar metal single-site catalysts, all of which also exhibit similar great 2e⁻ ORR performance including high H₂O₂ selectivity and large current density. However, the electrocatalyst from the present manuscript still shows unique significance and innovation due to its special satisfactorily competitive electrocatalytic performance associated with the good crystallinity and porosity of COF-based material. The highest FE_{H₂O₂} of CoPc-S-COF reaches 98 % at 0.63 V versus RHE (equivalent to an overpotential of 230 mV) in 1 M KOH at room temperature. CoPc-S-COF also maintains FE_{H₂O₂} >95% in the continuous H₂O₂ electroproduction for 20 h at a fixed current density of 125 mA cm⁻². The calculated FE_{H₂O₂} of CoPc-S-COF is close to CoPc-OCNT (FE_{H₂O₂} retains 96-100% in the continuous H₂O₂ electroproduction for 30 h at a fixed current density of 300 mA cm⁻²) from *Nat. Commun.* **14**, 172 (2023) and higher than Co HSACs from *Nat. Commun.* **14**, 1426 (2023) (retain FE_{H₂O₂} > 90% at 300 mA cm⁻² for at least 25 h) as well as CoPorBr/CNT from *Energy Environ. Sci.*, **16**, 446-459, (2023) (the highest FE_{H₂O₂} > 95% at an

overpotential of 200 mV). In addition, in all the references mentioned by this reviewer, carbon nanotubes (CNT) are necessary for preparing corresponding catalysts to improve the electron transport during the electrocatalytic process owing to the poor conductivity of the CoPc/CoPor molecules. However, CoPc-S-COF obtained in the present work exhibits competitive electrocatalytic performance without the help of any additional carbon materials due to the excellent electron transport capacity benefited from its long range ordered porous structure.

More impressively, we also focus on modifying the COF linkage to tuning its catalytic performance as also mentioned by this reviewer. To further enhance the understanding of the performance improvement in the present case, we have carried out a series of theoretical calculations. The adsorption energy (ΔG_{ads}) of O_2 on various atoms of CoPc-S-COF and CoPc-O-COF including Co, N, O, S, and F has been calculated to explore the active site of H_2O_2 production. The significantly smaller value of ΔG_{ads} for Co atom, *ca.* -0.3 eV, than those for other atoms (-0.02 ~ -0.05 eV for N, +0.4 ~ +0.6 eV for O/S/F) reveals the active site nature of Co atom in these two COFs towards ORR. Moreover, the electron density on the CoPc moiety including around Co site in CoPc-S-COF is higher than that for CoPc-O-COF due to the electron-donating effect of S atoms, which facilitates the charge transfer between Co active sites and intermediates and in turn affords enhanced catalytic activity for $2e^-$ ORR. In addition, the natural population analysis (NPA) charge distribution calculation for both COFs was also carried out to assess the atomic charge states in both COFs. According to the results displayed in Supplementary Table 6 (actually Table R1 also given below), the change of atomic charges on the conjugated chain is in a wave manner of $\text{S}\uparrow\text{-C}_a\downarrow\text{-C}_b\uparrow\text{-C}_c\downarrow\text{-C}_d\uparrow\text{-(NCo)}\downarrow$ during the electron transferring process from S/O atoms to central Co atoms, exhibiting an alternative polarization effect along the π electron transferring pathway in both COFs. In particular, the central Co atom gains an additional charge of $\sim 30 \times |10^{-3}e|$ when the O atoms in CoPc-O-COF are replaced by S atoms in CoPc-S-COF, exhibiting more obvious charge transfer from S to Co compared to that from O to Co. Furthermore, the calculated projected density of states (pDOS) discloses a lower d band center position and higher intensity of peaks near the Fermi level (E_f) for CoPc-S-COF compared to those of CoPc-O-COF, Fig. 3h. This indicates that larger density of active electrons around Co centers in CoPc-S-COF participates in the electrochemical ORR reaction, confirming the higher catalytic activity of CoPc-S-COF due to the electron-donating effect of S atoms. Nevertheless, Fig. 3i presents the calculated Gibbs free energy differences (ΔG) diagrams of the $2e^-$ ORR and $4e^-$ ORR processes on CoPc-S-COF and CoPc-O-COF. As can be found, the weaker $^*\text{OOH}$ adsorption and stronger $^*\text{H}_2\text{O}_2$ adsorption on CoPc-S-COF compared to CoPc-O-COF lead to a lower OOH^* to H_2O_2 conversion energy barrier of 0.63 eV for CoPc-S-COF than that for CoPc-O-COF, 1.11 eV, indicating the enhanced $2e^-$ ORR activity of the former species over the latter one. It is worth noting that the weaker $^*\text{OOH}$ adsorption on this electron-rich Co site in CoPc-S-COF is also verified by coresponding NPA charge distribution calculation as mentioned above. As shown in Supplementary Table 6 and Fig. 46 (actually Table R1 and Figure R1 also given below), the atomic charge of the inner N atoms is enhanced

from $-320 \times |10^{-3}e|$ to $-373 \times |10^{-3}e|$ when the O atoms in CoPc-O-COF are replaced by S atoms in CoPc-S-COF. Due to the increased negative charge located at the inner N atoms, the total repulsive force between O_a - O_b in *OOH and inner- N_4 framework in Pc macrocycle is as high as $0.076 e^2 \text{ Bohr}^{-2}$ in CoPc-S-COF, which is obviously larger than that in CoPc-O-COF of $0.057 e^2 \text{ Bohr}^{-2}$. As a result, the repulsive force between *OOH and N_4 gets enhanced, which is responsible for the weakened Co-OOH bonding in electron-rich Co site of CoPc-S-COF. Actually, these calculation findings are also in line with the results reported previously for the Pc/Por-based catalysts, see for examples: *CCS Chemistry* **4**, 228-236 (2021) and *Energy Environ. Sci.*, **16**, 446-459 (2023).

Fig. R1. The schematic drawing of enhanced repulsive force between *OOH and N_4 .

Table R1. The atomic charges ($|e|$) of CoPc-S-COF and CoPc-O-COF.

	S/O	C _a	C _b	C _c	C _d	N	Co
 CoPc-S-COF	+0.214	-0.118	+0.115	-0.051	+0.191	-0.373	+0.791
 CoPc-O-COF	-0.164	+0.068	+0.090	-0.024	+0.169	-0.320	+0.820
Difference between CoPc-S-COF and CoPc-O-COF.	0.378	-0.186	0.025	-0.030	0.022	-0.053	-0.029

Reviewer #2 (Remarks to the Author):

In this manuscript, the authors reported a bridge bonded metal phthalocyanine framework with undulated layers for H₂O₂ electrosynthesis. This work is interesting in crosslinking CoPc and COF, but the current manuscript cannot be accepted unless the following issues have been soundly addressed.

Answer: We really appreciate this reviewer's valuable comments and suggestions towards enhancing the quality of our research. The responses to the specific comments are detailed below.

1. The supplementary Fig. 29 didn't match the description in main text.

Answer: Thanks a lot for this suggestion! The accurate legend has been updated in Supplementary Fig. 30 of revised version of supplementary information (actually Fig. R2 also given below).

Fig. R2. ORR polarization curves in 0.1 M KOH before and after the addition of 1 mM SCN^- .

2. In Figure 4c, the current density of the ordinate unit was marked as A cm^{-2} , which was described as mA cm^{-2} in discussion part. The same mistake could also be observed in supplementary Fig. 39.

Answer: Thanks a lot for the reviewer's careful comment! As a result, the current density of the ordinate unit in Figure 4c and Supplementary Fig. 40 in the revised version of manuscript and supplementary information have been corrected as mA cm^{-2} according to this instruction.

3. As is known to all, the peak intensity of R space curves directly associates with the coordination number of Co. In supplementary Fig. 17, the peak intensity of CoPc with Co-N4 configuration is significantly higher than that of CoPc-O-COF and CoPc-S-COF, which demonstrates that the coordination numbers of Co-N for CoPc-O-COF and CoPc-S-COF are lower than that of CoPc. Besides, the EXAFS data should be fitted based on standard sample (e.g. CoPc in this work), but the authors only displays the EXAFS fitting results of CoPc-O-COF and CoPc-S-COF. In addition, the valence states obtained from XANES spectra should compared by standard samples. In this work, charge transfer was obviously observed from the XPS results (Fig. 2g), which may also lead to a significant change in XANES spectra. Hence, the data processing of the XAFS in this manuscript is unprofessional and insufficient to fully reveal the local structure information of the material.

Answer: Thanks a lot for this comment and suggestion! According to the instruction, EXAFS fitting results of both COFs were re-fitted based on standard sample CoPc, and corresponding results have been updated in Supplementary Fig. 18 and Table 3 in the revised version of supplementary information (actually Fig. R3 also given below). It turns out that both COFs and standard CoPc exhibit similar peak at 1.4 Å due to the

Co-N scattering path. In addition, the EXAFS fitting results of standard CoPc show a coordination number of 4, which is similar to both COFs.

Fig. R3. (a) Fourier transformation of EXAFS spectra of CoPc, Co foil, CoPc-O-COF, and CoPc-S-COF. (b) The EXAFS fitting results of CoPc-O-COF, CoPc-S-COF and CoPc.

In addition, the more detail of valence states obtained from XANES spectra is also added in the Supplementary Fig. 17 of revised version of supplementary information (actually Fig. R4 also given below).

Fig. R4. Co K-edge XANES spectra of CoPc, CoPc-O-COF, CoPc-S-COF, and Co foil.

4. The Ag/AgCl reference electrode is not suitable for alkaline conditions, Hg/HgO reference electrode is suggested.

Answer: Thanks a lot for this comment and suggestion! It is well known that in addition to the Hg/HgO reference electrode, actually the Ag/AgCl reference electrode has also been widely utilized in the alkaline electrolyte due to its low cost and low hypotoxicity, see for examples: *Adv. Sci.* **9**, 2104768 (2022); *J. Am. Chem. Soc.* **142**, 8104–8108

(2020); *Nat. Commun.* **14**, 172 (2023). Nevertheless, the potential value of Ag/AgCl reference electrode was calibrated before tests to ensure the reliability of the experimental data in the alkaline condition.

5. In line 234 to 236, the author declares that the higher double layer capacitances of CoPc-O-COF and CoPc-S-COF in turn leading to its higher 2e⁻ ORR performance. How about the relationship between double layer capacitances and ORR selectivity?

Answer: Good question and thanks a lot for this comment! It is well known that the electrochemically active surface areas (ECSAs) of electrocatalysts have been evaluated based on the electrochemical double layer capacitance (C_{dl}) using the cyclic voltammetry method, which is usually used to assess the extent of active site exposure during electrocatalytic process, see for examples: *Adv. Sci.* **9**, 2104768 (2022); *J. Am. Chem. Soc.* **142**, 8104–8108 (2020). In the revised version manuscript, the larger C_{dl} value of CoPc-S-COF (241 $\mu\text{F cm}^{-2}$) than that of CoPc-O-COF (171 $\mu\text{F cm}^{-2}$) indicates more available active sites within CoPc-S-COF originated from its twisted layered structure, leading to its higher 2e⁻ ORR performance.

6. As illustrated in Fig 3g, the oxygen and sulfur atoms are far away from the cobalt atom, and this electron contribution effect may be diluted by the surrounding carbon and nitrogen atoms, thereby resulting in an insufficiently defined electronic structure for cobalt.

Answer: Good question and thanks a lot for this comment! It is well known that the electron-donating/-withdrawing nature of substituents could be divided into the long-range conjugated effect and short-range inductive effect, see for examples: *Organic Chemistry*, 8th ed., Solomons, T. W. G., Fryhle, C. B., Eds., John Wiley & Sons, Inc., **2004**, pp 112–113; *Advanced Organic Chemistry (PartA)*, 5th ed., Carey, F. A., Sundberg, R. J., Eds., Springer-Verlag: Berlin, **2007**, pp12, 338. In CoPc-S-COF, the uniform conjugated system constructed by the benzene linkage, S atoms, and CoPc moieties forms a good pathway to exhibit the long-range conjugated effect. In order to visualize the uniform conjugated system, the localized orbital locator for π electrons (π -LOL) calculation is carried out using ω B97X-D functional in CoPc-S-COF, Supplementary Fig. 45 (actually Fig. R5 also given below). As can be found, the S atoms have been completely fused into the uniform conjugated system using the π -bonding in the manner of $\pi(\text{PcCo})-\text{p}(\text{S})-\pi(\text{Ph})$. Follow this long-range π linking pathway, the charge transfer from S to Co is quite smooth due to the long-range conjugated effect. This is also true for CoPc-O-COF. In addition, the NPA charge distribution calculation gives the result that the charges of adjacent atoms are positive or negative during the electron transferring process from S/O atoms to central Co atoms, indicating an alternative polarization effect along the π electron transferring pathway in both COFs. In particular, the central Co atom gains additional charge of $\sim 30 \times |10^{-3}e|$ when the O atoms in CoPc-O-COF are replaced by S atoms in CoPc-S-COF, exhibiting more obvious charge transfer from S to Co compared to that from O to Co.

Fig. R5. (a) The visualized scheme of the uniform conjugated system of CoPc-S-COF (top) and CoPc-O-COF (bottom). The schematic diagram of the uniform conjugated system in top/side/bottom view for (b) CoPc-S-COF and (c) CoPc-O-COF.

Reviewer #3 (Remarks to the Author):

Manuscript: NCOMMS-23-34167

"Dithiine-linked Metalphthalocyanine Framework with Undulated Layers for Highly Efficient and Stable H₂O₂ Electroproduction", by Qianjun Zhi, Rong Jiang, Xiya Yang, Yucheng Jin, Dongdong Qi, Kang Wang, and Jianzhuang Jiang

In this work, cobalt phthalocyanine (CoPc)-based covalent organic frameworks (COF) are prepared and used for the two-electron oxygen reduction reaction (ORR) to generate hydrogen peroxide. This is a thorough study in which various physico-chemical characterization methods are employed. The results of the electrochemical investigations are reliable. There are still some issues that need to be addressed.

Overall. In current form the paper is not suitable for publication in this highly ranked Journal. Recommendation: major revision

Answer: We indeed appreciate this reviewer's affirmation regarding our work and the valuable suggestions towards further improving the quality of the present research. The responses to the specific comments are detailed below.

Comments to the authors:

1. Page 3, lines 41-43. "As a consequence..." This sentence needs a reference to be made (see, for example, Lobyntseva et al., doi:10.1016/j.electacta.2007.05.076). This paper should be cited.

Answer: Thanks a lot for this suggestion! We have added the reference (Lobyntseva et

al., doi:10.1016/j.electacta.2007.05.076) as Ref. 14 in the revised version of manuscript.

2. Page 3, line 49. *This Journal has a wide readership. It would be of particular interest for the readers of this Journal to provide the main types of electrocatalysts used for hydrogen peroxide electrosynthesis. Literature should be consulted in this regard.*

Answer: Thanks a lot for this suggestion! We have provided some references on the main types of electrocatalysts used for hydrogen peroxide electrosynthesis including modified carbon, noble-metal and alloys, none-noble metals, metal-organic frameworks (MOFs), and covalent organic frameworks (COFs)-based electrocatalysts, which are also added as Ref. 15-19 in the revised version of manuscript.

3. Page 7, line 164. *What could be the reason for a large difference between the BET values of CoPc-O-COF and CoPc-S-COF materials? Pore size distribution should be added to Supplementary Fig. 13.*

Answer: Thanks a lot for this comment and question! According to the instruction, the pore size distribution of both COFs has been added in Supplementary Fig. 14 in the revised version of supplementary information (actually Fig. R6 also given below). The calculated pore volume and pore size distribution are $0.09 \text{ cm}^3 \text{ g}^{-1}$ & 1.7 nm for CoPc-O-COF and $0.14 \text{ cm}^3 \text{ g}^{-1}$ & 1.5 nm for CoPc-S-COF. Although the pore size of CoPc-S-COF is smaller, it has a larger total pore volume than that of CoPc-O-COF, due to the larger undulated layer spacing of CoPc-S-COF originated from its C-S-C linking unit, thus leading to a larger BET surface area of CoPc-S-COF.

Fig. R6 Pore volume and pore size distribution of (a) CoPc-O-COF and (b) CoPc-S-COF.

4. Page 8, line 202. *At which current density threshold the ORR onset potential was determined?*

Answer: Thanks a lot for this question! The onset potentials of all samples were defined at -0.1 mA cm^{-2} of H_2O_2 partial current according to corresponding references, see for

examples: *Nat. Commun.* **14**, 172 (2023) and *Nat. Commun.* **12**, 4225 (2021), which have been added as Ref. 20 and 49 in the revised version of manuscript.

5. Page 8, last sentence. The Tafel analysis is also used to discuss about the ORR mechanism on the catalyst materials. The discussion about the mechanism of the ORR should be added.

Answer: Thanks for this good suggestion! According to the instruction, the Tafel slope of individual CoPcF₁₆ has also been added in Supplementary Fig. 19 of revised version of supplementary information (actually Fig. R7 also given below). Moreover, the Tafel slopes of CoPc-O-COF and CoPc-S-COF are calculated to be 49 and 47 mV dec⁻¹, respectively, smaller than that of individual CoPcF₁₆ (76 mV dec⁻¹). This indicates the faster 2e⁻ ORR kinetics of both COFs, which might be attributed to their high initial electron transfer efficiency and large active surface during the catalytic process.

Fig. R7 Tafel plots of CoPc-O-COF, CoPc-S-COF, and CoPcF₁₆.

6. Page 10, lines 225-227. The mass activity (MA) for ORR might depend on the catalyst loading. As usual, the MA values are calculated in the kinetic region of the ORR process. It is pointless to calculate MA at 0.3 V in which the diffusion-limited current plateau is formed and the ORR process is controlled by mass transfer. This review suggests the authors to determine MA at 0.8 V. Therefore, the potential range in Supplementary Fig. 20 should be presented up to 0.75 V.

Answer: Thanks a lot for this good comment and suggestion! According to the instruction, we have changed the potential range to a more appropriate range of 0.7 to 1.0 V vs RHE in Supplementary Fig. 21 of the revised version of supplementary information (actually Fig. R8 also given below). In addition, the mass activity of CoPc-S-COF was determined as 38 A g_{Co}⁻¹ at 0.8 V vs RHE, superior to that of CoPc-O-COF, 25 A g_{Co}⁻¹ at 0.8 V vs RHE, which is also corrected in the revised version of manuscript.

Fig. R8 Mass activity of CoPc-O-COF and CoPc-S-COF.

7. Page 10, line 239. Why is the active amount of CoPc units so low (only 11.0%)?

Answer: Good question! Referring to the results reported previously in *J. Am. Chem. Soc.* **139**, 4258–4261 (2017); *Nat. Commun.* **9**, 576 (2018); *Nat. Commun.* **11**, 497, (2020), 2D extended polygons tend to pack closely in an eclipsed fashion in COFs due to the strong π - π interactions, the interior active sites buried deeply inside the 1D channels are therefore difficult to be reached due to the long ion transportation path, which will inevitably lead to insufficient utilization of the active sites and result in low electrocatalysis performance. Generally, the utilization of active sites of the 2D eclipsed COF electrocatalysts reported previously are usually less than 8%, see for examples: *Science* **349**, 1208–1213 (2015); *Angew. Chem. Int. Ed.* **61**, e202114244, (2022); *Nat. Commun.* **11**, 497, (2020). This actually is also true for the CoPc-O-COF catalyst in the present case, with only 6.9 % active atom utilization. Moreover, 11.0% active atom utilization in CoPc-S-COF is not only the evidence of more exposed active sites than CoPc-O-COF, but also represents a high level of active atom utilization for the 2D COF electrocatalysts reported thus far.

8. Page 11, lines 259-264 (Supplementary Fig. 29). The poisoning tests were performed in the presence of 1 mM SCN⁻. One can assume that Co-N₄ sites are blocked by adding SCN⁻ anions. However, still a substantial ORR current is observed in the presence of 1 mM SCN⁻. What could be those active sites, which catalyze the ORR in 0.1 M KOH containing 1 mM SCN⁻?

Answer: Thanks a lot for this good question and comment! The poison experiment was carried out to identify the catalytic site by using SCN⁻ ions, which tend to bind the metal atoms of CoPc and block the adsorption of reaction intermediates, thus negatively affecting the catalytic performance. As shown in Supplementary Fig. 30 in the revised version of supplementary information (actually Fig. R2 also given above), significant degradation of the catalytic activity occurs after adding 1 mM SCN⁻ into the electrolyte,

demonstrating the actual active site nature of the Co metal center for $2e^-$ ORR in both CoPc-O-COF and CoPc-S-COF. A substantial ORR current can still be observed in the presence of 1 mM SCN^- for both COFs owing to that not all Co metal centers are poisoned in 1 mM SCN^- . This can be further confirmed by the further degradation of ORR currents for both COFs along with the increase in the concentration of SCN^- from 1 to 25 mM, Fig. R9 given below.

Fig. R9 ORR polarization curves in 0.1 M KOH before and after the addition of 1 and 25 mM SCN^- .

9. Page 12, lines 291-293 (Supplementary Fig. 32). What could be the reason for a different ORR activity between CoPc-O and CoPc-S? In Figure 1a CoPc-O-COF and CoPc-S-COF catalysts show rather similar ORR electrocatalytic activity.

Answer: Thanks for this good question and comment! The two small molecule Co phthalocyanines including CoPc-S and CoPc-O containing C-S-C and C-O-C groups, respectively, were synthesized with their $2e^-$ ORR performance assessed to further clarify the effect of S and O atoms on the catalytic performance of the Co active center. On the basis of experimental result, CoPc-S electrode exhibits better $2e^-$ ORR activity and worse H_2O_2RR activity in comparison with CoPc-O, Supplementary Figs. 33 and 34 in the revised version of supplementary information, proving the effect of S on enhancing the $2e^-$ ORR performance of CoPc-S-COF. It is worth noting that the non-planar molecular structure of CoPc-S weakens the intermolecular aggregation, leading to better dispersion of CoPc-S in catalyst ink during the preparation of the working electrode compared to CoPc-O. This in turn results in a more different ORR activity between CoPc-O and CoPc-S compared to that between CoPc-O-COF and CoPc-S-COF.

10. Page 14, lines 360-362. The authors should perform a thorough physical characterization (post-mortem analysis) of the catalyst materials after soaking them in a peroxide solution for longer time to check the durability of these catalysts in a higher

concentration of H_2O_2 solution.

Answer: Thanks a lot for this comment and suggestion! According to the instruction, both CoPc-S-COF and CoPc-O-COF were immersed into 3% H_2O_2 solution for three days. As shown in Supplementary Figs. 7 and 8 in the revised version of supplementary information (actually Fig. R10 also given below), the PXRD patterns of both COFs recollected after immersion treatments remain almost unchanged, proving the durability of these two COFs.

Fig. R10 PXRD patterns of (a) CoPc-S-COF and (b) CoPc-O-COF after immersion in different solutions.

Minor remarks:

Page 3, line 44. ‘electrocatalytic’ should be ‘electrocatalytic’ (see also page 12, line 296)

Answer: Thanks a lot for this comment! According to the instruction, “electrocatalytic” has been corrected as “electrocatalytic” in the revised version of manuscript.

Page 4, line 97. ‘trimethylamine (Et3N)’ This is not correct.

Answer: Thanks a lot for this comment! According to the instruction, “trimethylamine” has been corrected as “triethylamine” in the revised version of manuscript.

Page 8, line 176. “shift to a lower energy” should be “shift to a lower binding energy”

Answer: Thanks a lot for this comment! According to the instruction, “shift to a lower energy” has been corrected as “shift to a lower binding energy” in the revised version of manuscript.

Page 8, line 202. ‘high reactivity’ should be replaced with ‘high electrocatalytic activity’

Answer: Thanks a lot for this comment! According to the instruction, “high reactivity” has been replaced with “high electrocatalytic activity” in the revised version of manuscript.

Page 10, line 223. “(K-L) diffusion equation” should be replaced with “(K-L) equation”

Answer: Thanks a lot for this comment! According to the instruction, “(K-L) diffusion equation” has been replaced with “(K-L) equation” in the revised version of manuscript.

Page 14, line 346. “excellent catalytic activity” should be “excellent electrocatalytic activity”

Answer: Thanks a lot for this comment! According to the instruction, “excellent catalytic activity” has been corrected as “excellent electrocatalytic activity” in the revised version of manuscript.

Supplementary material, page 4, line 79. KV should be kV as the unit of voltage

Answer: Thanks a lot for this comment! According to the instruction, “KV” has been corrected as “kV” in the revised version of supplementary information.

Supplementary material, page 6, line 130. “to respond to” should be “to detect”

Answer: Thanks a lot for this comment! According to the instruction, “to respond to” has been replaced with “to detect” in the revised version of supplementary information.

Supplementary material, page 9, line 213. ‘chronopotentiometry’ should be ‘chronoamperometry’

Answer: Thanks a lot for this comment! According to the instruction, “chronopotentiometry” has been corrected as “chronoamperometry” in the revised version of manuscript.

REVIEWER COMMENTS

Reviewer #2 (Remarks to the Author):

Most of my questions have been addressed, but the following issues should pay more attentions:

(1) Although Ag/AgCl reference electrode is used in alkaline conditions in many top publications, the alkaline condition may shorten the service life of Ag/AgCl electrode due to the generation of AgOH. Considering the cost and toxicity, a salt bridge is suggested when use AgAgCl in alkaline electrolytes.

(2) The authors should offer more details about how the peak intensities of CoPc-S-COF and CoPc-O-COF displayed in R space have increased to the same to CoPc. Please also supplement the K space spectra and indicate the K range in this manuscript when fitting the EXAFS data.

(3) Additionally, the EXAFS data were fitted after stability test, but no fitting results could be observed. Besides, K-space spectra is also suggested to displayed in Figure S29.

Reviewer #3 (Remarks to the Author):

Manuscript: NCOMMS-23-34167A

"Dithiine-linked Metalphthalocyanine Framework with Undulated Layers for Highly Efficient and Stable H₂O₂ Electroproduction", by Qianjun Zhi, Rong Jiang, Xiya Yang, Yucheng Jin, Dongdong Qi, Kang Wang, and Jianzhuang Jiang

In my opinion the authors satisfactory answered most of the reviewers' comments and questions. However, there are still some issues that need to be addressed. Recommendation: minor revision

Further comments to the authors:

1. Page 3, line 45. Quinone-modified carbon electrodes have been also employed for hydrogen peroxide electrosynthesis in alkaline media (see for example papers by Schiffrin and co-workers: J. Electroanal. Chem. 515, 101-112 (2001); J. Electroanal. Chem. 541, 23-29 (2003); J. Electroanal. Chem. 564, 159-166 (2004); Electrochim. Solid-State Lett. 8, E30-E33 (2005); Electrochim. Acta 50, 5126-5131 (2005). These papers are relevant in the context of this work.

2. Page 5, line 113. A thorough physical characterization of the catalyst materials should be made after soaking into 3% H₂O₂ solution for three days. PXRD alone is not sufficient for this analysis.
3. Page 7, line 167. 'pore size distribution' is misleading here. Average pore size should be determined.
4. Page 11, lines 270-272. The authors reply is not sufficient to the reviewer's previous comment #8. It is clear that SCN- poisoning blocks Co-N₄ active sites. However, there is still substantial ORR activity remaining. What could be the reason for that?

Minor remarks:

Page 3, line 43. 'none-noble' should be 'non-noble'

Page 4, line 75. 'metallic phthalocyanine' should be 'metal phthalocyanine'

Page 4, line 78. 'CoPc' should be replaced with 'cobalt phthalocyanine (CoPc)'

Page 8, line 193, 'pre-dge' should be 'pre-edge'

Page 10, line 227. 'voltage range' should be 'potential range' (see also page 10, line 230; page 12, line 293)

Reviewer #4 (Remarks to the Author):

Zhi et al. reported the Dithiine-linked CoPc framework for electrosynthesis the hydrogen peroxide. They have demonstrated their catalysts with higher selectivity and activities. However, I cannot recommend this manuscript for publishing at the Nature Communications, caused by the following points.

1. In abstracts: "Realization of stable and industrial-level H₂O₂ electroproduction still faces great challenge due large partly to the easy decomposition of H₂O₂." The full work implemented at the alkaline environments; I cannot believe their invents avoid the decomposition of H₂O₂.
2. The authors showed explain why the dithiine-linked CoPc is undulated layers.
3. The authors showed provide the error bar of EXAFS fitting. More importantly, the fitting of high shell (Co-C) is also necessary. The EXAFS fitting parameters of Supplementary fig 29 were disappeared.
4. The EXAFS fitting was wrong. In EXAFS fitting, CoPc-S-COF showed the low σ^2 smaller than 3.0×10^{-3} . It was not followed treaty for EXAFS fitting. From CoPc to CoPc-O-COF, the Co-N length were from 1.92 to 1.89 Å, it seems the high difference.

5. According to XPS and K-edge XAS, I cannot acknowledge your demonstration for electron transfer to Co centers. The authors should collect the L-edge XAS of the samples, and calculated the XAS curves.

Reviewer #2 (Remarks to the Author):

Most of my questions have been addressed, but the following issues should pay more attentions:

Answer: We really appreciate this reviewer's affirmation. The responses to the following valuable comments and suggestions towards further enhancing the quality of our research are detailed below.

(1) Although Ag/AgCl reference electrode is used in alkaline conditions in many top publications, the alkaline condition may shorten the service life of Ag/AgCl electrode due to the generation of AgOH. Considering the cost and toxicity, a salt bridge is suggested when use Ag/AgCl in alkaline electrolytes.

Answer: Thanks a lot for this professional comment and suggestion! Accordingly, all the electrochemical measurements in alkaline electrolyte have been performed by using Ag/AgCl reference electrode with a salt bridge. Corresponding data have been updated in the revised discussion, Fig. 3 and Supplementary Figs. 24-28, 31, 35, and 36-42 in the revised version of manuscript and Supplementary Information. It is worth noting that the ORR performance of all the catalysts in this work by using Ag/AgCl reference electrode with or without a salt bridge are very similar, as reflected in the slightly smaller starting potential (0.81V vs RHE) of CoPc-S-COF and nearly same H₂O₂ selectivity revealed by using Ag/AgCl reference electrodes with or without a salt bridge, Fig. R1 given below.

Fig. R1. (a) LSVs of CoPc-S-COF at 1600 rpm in O₂-saturated 0.1 M KOH with or without salt bridge. (b) H₂O₂ selectivity and electron transfer number n of CoPc-S-COF with or without salt bridge.

(2) The authors should offer more details about how the peak intensities of CoPc-S-COF and CoPc-O-COF displayed in R space have increased to the same to CoPc. Please also supplement the K space spectra and indicate the K range in this manuscript when fitting the EXAFS data.

Answer: Thanks a lot for this comment and suggestion. Theoretically, the peak intensity in R space is mainly related to the values of coordination number and Debye-Waller factor (σ^2). Besides, data collected over different periods and the signal-noise-ratio (SNR) of data can

also skew the results. In our previous experiment, the XAFS spectra of the standard CoPc and the synthesized samples (CoPc-S-COF and CoPc-O-COF) were collected at different times, and the integration time in a step-by-step scanning of the monochromator was relatively short, which caused a low SNR data that affected the fitting results. To ensure the reliability of the results, we re-collected the XAFS data of the three samples using a relatively long integration time under the same experimental conditions in the same period. The Fourier Transform EXAFS spectra (R-space) of the three samples on the basis of the newly collected XAFS data are shown in Supplementary Fig. 22 in the revised version of Supplementary Information (actually Fig. R2 also given below), indicating almost same intensity in the first shell of R-space. In addition, the corresponding k-space spectra and the EXAFS fitting parameters are provided in Supplementary Fig. 23 and Supplementary Table 4 in the revised version of Supplementary Information (actually Fig. R3 and Table R1 also given below). According to the EXAFS fitting results, the k values are determined to be in the range of 2.5~11 Å⁻¹ with a weight of 3.

Fig. R2. (a) Fourier transform of EXAFS spectra of CoPc, Co foil, CoPc-O-COF, and CoPc-S-COF. (b) The EXAFS fitting results of CoPc-O-COF, CoPc-S-COF and CoPc.

Fig. R3. The Co k^3 -weighted K -space spectra of CoPc-O-COF, CoPc-S-COF, and CoPc.

Table R1. The fitting parameters of the Fourier transform EXAFS spectra.

Sample	Bond	N	R(Å)	σ^2 (10^{-3}Å^2)	ΔE_0 (eV)	R factor
CoPc-S-COF (before tests)	Co-N (first shell)	4	1.91 ± 0.01	4.2 ± 0.4	2.5 ± 1.4	0.002
	Co-C (second shell)	8	2.99 ± 0.01	7.1 ± 0.5	3.5 ± 1.0	
	Co-N (third shell)	4	3.26 ± 0.01	7.1 ± 0.5	3.5 ± 1.0	
CoPc-S-COF (after tests)	Co-N (first shell)	4	1.92 ± 0.02	8.7 ± 1.1	7.9 ± 3.6	0.003
	Co-C (second shell)	8	2.99 ± 0.03	8.3 ± 1.2	5.5 ± 3.7	
	Co-N (third shell)	4	3.29 ± 0.03	8.3 ± 1.2	5.5 ± 3.7	
CoPc-O-COF (before tests)	Co-N (first shell)	4	1.92 ± 0.01	4.3 ± 1.2	7.1 ± 3.9	0.015
	Co-C (second shell)	8	3.00 ± 0.02	7.5 ± 1.4	4.6 ± 2.5	
	Co-N (third shell)	4	3.27 ± 0.03	7.5 ± 1.4	4.6 ± 2.5	
CoPc-O-COF (after tests)	Co-N (first shell)	4	1.90 ± 0.02	7.0 ± 1.2	4.0 ± 2.4	0.010
	Co-C (second shell)	8	2.98 ± 0.03	9.0 ± 1.8	5.9 ± 2.7	
	Co-N (third shell)	4	3.27 ± 0.03	9.0 ± 1.8	5.9 ± 2.7	
CoPc	Co-N (first shell)	4	1.90 ± 0.02	7.0 ± 1.2	7.0 ± 2.6	0.017
	Co-C (second shell)	8	2.98 ± 0.03	9.0 ± 1.8	6.6 ± 3.5	
	Co-N (third shell)	4	3.27 ± 0.03	9.0 ± 1.8	6.6 ± 3.5	

(3) Additionally, the EXAFS data were fitted after stability test, but no fitting results could be observed. Besides, K-space spectra is also suggested to displayed in Figure S29.

Answer: Thanks a lot for this comment and suggestion. Accordingly, the fitting results and K-space spectra of the catalysts CoPc-O-COF and CoPc-S-COF after stability test have been added into the Supplementary Table 4 and Supplementary Fig. 34 in revised version of supplementary information (actually Table R1 and Fig. R4 also given above and below,

respectively).

Fig. R4. The Co K-edge XANES spectra of (a) CoPc-O-COF and (d) CoPc-S-COF before and after stability tests. The EXAFS fitting results of (b) CoPc-O-COF and (e) CoPc-S-COF before and after stability tests. The Co k^3 -weighted K -space spectra of (c) CoPc-O-COF and (f) CoPc-S-COF before and after stability tests.

Reviewer #3 (Remarks to the Author):

Manuscript: NCOMMS-23-34167A

"Dithiine-linked Metalphthalocyanine Framework with Undulated Layers for Highly Efficient and Stable H₂O₂ Electroproduction", by Qianjun Zhi, Rong Jiang, Xiya Yang, Yucheng Jin, Dongdong Qi, Kang Wang, and Jianzhuang Jiang

In my opinion the authors satisfactory answered most of the reviewers' comments and questions. However, there are still some issues that need to be addressed. Recommendation: minor revision.

Answer: We indeed appreciate this reviewer's affirmation regarding our work and the valuable suggestions towards further improving the quality of the present research. The responses to the specific comments are detailed below.

Further comments to the authors:

1. Page 3, line 45. Quinone-modified carbon electrodes have been also employed for hydrogen peroxide electrosynthesis in alkaline media (see for example papers by Schiffrin and co-workers: J. Electroanal. Chem. 515, 101-112 (2001); J. Electroanal. Chem. 541, 23-29 (2003); J. Electroanal. Chem. 564, 159-166 (2004); Electrochem. Solid-State Lett. 8, E30-E33 (2005); Electrochim. Acta 50, 5126-5131 (2005). These papers are relevant in the context of this work.

Answer: Thanks a lot for this suggestion! Accordingly, all these references have been added

as Ref. 16-20 in the revised version of manuscript.

2. Page 5, line 113. A thorough physical characterization of the catalyst materials should be made after soaking into 3% H₂O₂ solution for three days. PXRD alone is not sufficient for this analysis.

Answer: Thanks a lot for this comment and suggestion! The FTIR spectra and TEM images of CoPc-O-COF and CoPc-S-COF after being soaked into 3% H₂O₂ solution for three days were recorded to further assess the stability of these catalysts in H₂O₂ solution. As shown in Supplementary Figs. 9 and 10 in the revised version of Supplementary Information (actually Figs. R5 and R6 also given below), both CoPc-O-COF and CoPc-S-COF display very similar FTIR spectra and TEM images before and after the soaking treatment, further proving the durability of CoPc-O-COF and CoPc-S-COF in 3% H₂O₂ solution.

Fig. R5. TEM and HR-TEM images of (a, b, e, f) CoPc-O-COF and (c, d, g, h) CoPc-S-COF before and after soaking into 3% H₂O₂ solution for three days.

Fig. R6. FT-IR spectra of (a) CoPc-O-COF and (b) CoPc-S-COF before and after soaking into 3% H₂O₂ solution for three days.

3. Page 7, line 167. ‘pore size distribution’ is misleading here. Average pore size should be determined.

Answer: Thanks a lot for this comment and suggestion! N₂ adsorption-desorption measurements reveal the pore size distribution of 1.5-2.3 and 1.3-2.1 nm for CoPc-O-COF and CoPc-S-COF, respectively, with an average pore size of 1.7 and 1.5 nm. Corresponding discussion has been modified on Page 7, line 167 in the revised version of manuscript.

4. Page 11, lines 270-272. The authors reply is not sufficient to the reviewer’s previous comment #8. It is clear that SCN⁻ poisoning blocks Co-N₄ active sites. However, there is still substantial ORR activity remaining. What could be the reason for that?

Answer: Thanks a lot for this comment! As shown in Fig. R7a, along with addition of 5 mM SCN⁻ into the electrolyte, the ORR current density of CoPc-S-COF and CoPc-O-COF at 0.6 V versus RHE gets reduced into -0.8 and -0.9 mA cm⁻² from the initial current density of -2.1 and -2.3 mA cm⁻². Further increasing the concentration of SCN⁻ to 25 mM does not lead to further significantly decrease in the ORR current density, revealing the almost complete poisoning of Co-N₄ active sites by SCN⁻. The remaining ORR current density could be derived from the ORR activity of the N-rich Pc macrocycle in CoPc-S-COF and CoPc-O-COF. To confirm this point, metal free H₂PcF₁₆ electrode was also prepared and tested on RRDE at 1600 rpm in O₂-saturated 0.1 M KOH. As can be seen in Fig. R7b given below, the H₂PcF₁₆ electrode also shows electrocatalytic ORR activity with a current density of -0.9 mA cm⁻² at 0.6 V versus RHE. These results indicate the fact that the remaining ORR activity of CoPc-S-COF and CoPc-O-COF after the SCN⁻ poisoning originates from the Pc macrocycles containing abundant N atoms and benzene moieties.

Fig. R7. (a) ORR polarization curves of CoPc-S-COF and CoPc-O-COF in 0.1 M KOH before and after the addition of 1, 5, and 25 mM SCN⁻. (b) ORR polarization curves of the metal free H₂PcF₁₆ electrode as well as CoPc-S-COF and CoPc-O-COF in 0.1 M KOH.

Minor remarks:

Page 3, line 43. ‘none-noble’ should be ‘non-noble’

Answer: Thanks a lot for this comment! According to the instruction, ‘none-noble’ has been corrected as ‘non-noble’ in the revised version of manuscript.

Page 4, line 75. 'metallic phthalocyanine' should be 'metal phthalocyanine'

Answer: Thanks a lot for this comment! According to the instruction, 'metallic phthalocyanine' has been changed into 'metal phthalocyanine' in the revised version of manuscript.

Page 4, line 78. 'CoPc' should be replaced with 'cobalt phthalocyanine (CoPc)'

Answer: Thanks a lot for this comment! According to the instruction, 'CoPc' has been replaced with 'cobalt phthalocyanine (CoPc)' in the revised version of manuscript.

Page 8, line 193, 'pre-dge' should be 'pre-edge'

Answer: Thanks a lot for this comment! According to the instruction, 'pre-dge' has been corrected as 'pre-edge' in the revised version of manuscript.

Page 10, line 227. 'voltage range' should be 'potential range' (see also page 10, line 230; page 12, line 293)

Answer: Thanks a lot for this comment! According to the instruction, 'voltage range' has been changed into 'potential range' in the revised version of manuscript.

Reviewer #4 (Remarks to the Author):

Zhi et al. reported the Dithiine-linked CoPc framework for electrosynthesis the hydrogen peroxide. They have demonstrated their catalysts with higher selectivity and activities. However, I cannot recommend this manuscript for publishing at the Nature Communications, caused by the following points.

Answer: First of all, we truly appreciate this reviewer's valuable comments towards improving the quality of the present research. The responses to the specific comments are detailed below.

1. In abstracts: "Realization of stable and industrial-level H₂O₂ electroproduction still faces great challenge due large partly to the easy decomposition of H₂O₂." The full work implemented at the alkaline environments; I cannot believe their invents avoid the decomposition of H₂O₂.

Answer: Thanks a lot for this professional comment! Indeed, H₂O₂ is more easily decomposed in the alkaline environment than in the neutral environment. However, the presence of metal ions (i.e. Mn, Fe, Co, and Pt) which usually act as active centers for the H₂O₂ electroproduction could also lead to the significant decomposition of H₂O₂, impeding the realization of stable and industrial-level H₂O₂ electroproduction. In this work, a dithiine-linked 2D CoPc-based COF, CoPc-S-COF, was afforded. The nonplanar dithiine groups lead to the undulated layered structure of CoPc-S-COF, which allows the exposition of more Co sites to enhance the catalytic performance. More importantly, the electronic effect of S atoms in the dithiine groups enables to activate the 2e⁻ ORR but deactivate the H₂O₂ decomposition capability of the Co center in CoPc-S-COF, which has been demonstrated by the higher electrocatalytic H₂O₂ production performance of CoPc-S-COF than that of the

dioxin-linked CoPc-O-COF tested in the alkaline electrolyte. To further reveal the application potential of CoPc-S-COF, the electrocatalytic H₂O₂ production activity of CoPc-S-COF has also been evaluated in 1 M Na₂SO₄ (pH = 7) in the gas diffusion electrode device. As shown in Supplementary Fig. 49 in the revised version of Supplementary Information (actually Fig. R8 also given below), CoPc-S-COF displays excellent 2e⁻ ORR activity in the neutral electrolyte with a high H₂O₂ selectivity (*ca.* 80%) in the potential range of 0.03 – 0.33 V versus RHE. More importantly, CoPc-S-COF displays a stable H₂O₂ production with a high FE_{H₂O₂} of *ca.* 80% under a current density of 100 mA cm⁻² at an overpotential of *ca.* 0.22 mV for 20 h, generating 256 mg H₂O₂ (corresponding to a H₂O₂ concentration of 1.53 wt%), comparable to the result obtained in alkaline electrolyte. These additional experimental results further confirm the outstanding electrocatalytic H₂O₂ production performance of CoPc-S-COF, which is beneficial to the design and preparation of high-performance and low-cost electrocatalysts towards industrial-level H₂O₂ electroproduction.

Fig. R8. H₂O₂ electroproduction in flow cell with 1 M Na₂SO₄ electrolyte. (a) LSV of CoPc-S-COF in flow cell. (b) The chronoamperometry measurements at varied applied voltages of CoPc-S-COF. (c) FE_{H₂O₂} of CoPc-S-COF at varied applied voltages. (e) H₂O₂ yields of CoPc-S-COF under a current density of 100 mA cm⁻². (d) Chronopotentiometry curve at a current density of 100 mA cm⁻² and the corresponding FE_{H₂O₂} in the flow cell for CoPc-S-COF.

2. The authors should explain why the dithiine-linked CoPc is undulated layers.

Answer: Thanks a lot for this comment! For the purpose of fixing this question, two model molecule compounds containing dithiine moieties have been synthesized, Supplementary Fig. 11 in the revised version of Supplementary Information (actually Fig. R9 also given below). On the basis of single crystal X-ray diffraction analysis result, the angle of the C-S-C bridge in the dithiine groups in the single crystals of these two model compounds amounts to 100.51° and 100.95°, respectively, leading to the V-shaped configuration of both model molecules. More importantly, such a V-shaped molecular structure of these two model compounds further induces the packing structure with an undulated arrangement in their single crystals, Supplementary Fig. 11 and Supplementary Table 1 in the revised version of

Supplementary Information (actually Fig. R9 and Table R2 given below). This is also true for CoPc-S-COFs according to their PXRD analysis result as detailed in Fig. 1 in the revised version of manuscript.

Fig. R9. (a, d) Schematic synthesis of the model molecules. (b, e) Molecular and (c, f) packing structure of the model molecules in their single crystals. C, H, and S are displayed in grey, white, and yellow, respectively.

Table R2. Single Crystal X-ray Diffraction Experimental Details

	Model molecule 1	Model molecule 2
CCDC number	2302889	2305173
Empirical formula	C ₁₂ H ₈ S ₂	C ₁₈ H ₁₀ S ₄
Formula weight	216.30	354.50
Temperature [K]	293(2)	293(2)
Crystal system	monoclinic	monoclinic
Space group (number)	P 2 ₁ / c (14)	C 2/ c
a [Å]	11.9449(8)	21.076(2)
b [Å]	6.1481(4)	10.0464(8)
c [Å]	14.4785(10)	7.4290(7)
α [°]	90	90
β [°]	109.998(8)	96.849(9)
γ [°]	90	90
Volume [Å ³]	999.17(13)	1561.8(2)
Z	4	4
ρ_{calc} [gcm ⁻³]	1.438	1.508
μ [mm ⁻¹]	0.483	0.600

$F(000)$	448	728
Radiation	Mo $K\alpha$ ($\lambda=0.71073$ Å)	Mo $K\alpha$ ($\lambda=0.71073$ Å)
2 θ range [°]	5.85 to 58.23 (0.73 Å)	4.50 to 58.43 (0.73 Å)
Index ranges	-11 $\leq h \leq$ 16 -8 $\leq k \leq$ 8 -18 $\leq l \leq$ 19	-28 $\leq h \leq$ 25 -12 $\leq k \leq$ 13 -10 $\leq l \leq$ 10
Reflections collected	4567	8492
Independent reflections	2249 $R_{\text{int}} = 0.0213$ $R_{\text{sigma}} = 0.0355$	1903 $R_{\text{int}} = 0.1091$ $R_{\text{sigma}} = 0.0752$
Completeness	98.7 %	99.9%
Data / Restraints / Parameters	2249/0/127	2249/0/100
Goodness-of-fit on F^2	1.054	1.028
Final R indexes [$I \geq 2\sigma(I)$]	$R_1 = 0.0354$ $wR_2 = 0.0822$	$R_1 = 0.0703$ $wR_2 = 0.1719$
Final R indexes [all data]	$R_1 = 0.0482$ $wR_2 = 0.0873$	$R_1 = 0.0937$ $wR_2 = 0.1890$
Largest peak/hole [$e\text{Å}^{-3}$]	0.20/-0.27	0.99/-0.31

3. The authors should provide the error bar of EXAFS fitting. More importantly, the fitting of high shell (Co-C) is also necessary. The EXAFS fitting parameters of Supplementary fig 29 were disappeared.

Answer: Thanks a lot for this comment and suggestion! As shown in Supplementary Table 4 in the revised version of the Supplementary Information, the error bar of EXAFS fitting parameters have been added. Moreover, the fitting of high shell was also carried out for CoPc, CoPc-O-COF, and CoPc-S-COF. As can be found in Supplementary Fig. 22 and Table 4 in the revised version of Supplementary Information (actually Fig. R2 and Table R1 also given above), the fitting results of all these materials show the peaks at *ca.* 1.53, 2.49, and 3.10 Å due to the Co-N (first shell), Co-C (second shell), and Co-N (third shell) scattering paths with a coordination number of 4, 8, and 4, agreeing well with the molecular structure of CoPc. In addition, the EXAFS fitting parameters of Supplementary Fig. 29 have also been provided in the Supplementary Table 4 in the revised version of Supplementary Information (actually Table R1 also given above).

4. The EXAFS fitting was wrong. In EXAFS fitting, CoPc-S-COF showed the low σ^2 smaller than 3.0×10^{-3} . It was not followed treaty for EXAFS fitting. From CoPc to CoPc-O-COF, the Co-N length were from 1.92 to 1.89 Å, it seems the high difference.

Answer: Thanks a lot for this professional comment. The small σ^2 of CoPc-S-COF is owing to the low signal-noise ratio of the EXAFS data. To obtain the satisfactory EXAFS fitting results, the XANES spectra of all the samples in this work have been re-recorded. As shown in Supplementary Table 4 in the revised version of Supplementary Information (actually Table R1 also given above), the EXAFS fitting results reveal the σ^2 value in the range of $4.2\text{-}9.9 \times 10^{-3}$ for all the samples on the basis of the newly recorded XANES spectra, in accordance with the treaty for EXAFS fitting. In addition, the new EXAFS fitting results also indicate a very similar Co-N (first shell) length for CoPc (1.90 ± 0.02 Å), CoPc-O-COF (1.92 ± 0.01 Å), and CoPc-S-COF (1.91 ± 0.01 Å).

5. According to XPS and K-edge XAS, I cannot acknowledge your demonstration for electron transfer to Co centers. The authors should collect the L-edge XAS of the samples, and calculated the XAS curves.

Answer: Thanks a lot for this comment and suggestion. Accordingly, the Co L_{2, 3}-edge XANES spectra of CoPc-O-COF and CoPc-S-COF were collected to further demonstrate the increased electron density of the Co center in CoPc-S-COF. As shown in Supplementary Fig. 21 in the revised version of Supplementary Information (actually Fig. R10 also given below), both L₂ and L₃-edges photon energies of CoPc-S-COF are 0.5 eV smaller than those of CoPc-O-COF, consisting with the results of XPS and K-edge XAFS analysis.

Fig. R10. Co L_{2, 3}-edge XANES spectra of CoPc-O-COF and CoPc-S-COF.

REVIEWER COMMENTS

Reviewer #2 (Remarks to the Author):

The current revision could be accepted.

Reviewer #3 (Remarks to the Author):

Manuscript: NCOMMS-23-34167B

"Dithiine-linked Metalphthalocyanine Framework with Undulated Layers for Highly Efficient and Stable H₂O₂ Electroproduction", by Qianjun Zhi, Rong Jiang, Xiya Yang, Yucheng Jin, Dongdong Qi, Kang Wang, Yunpeng Liu, and Jianzhuang Jiang

This manuscript could be accepted for publication in Nature Communications after minor revision.

Minor comments:

1. Page 12, lines 310-312. "However, the metal-free H₂PcF16 electrode displays much inferior electrocatalytic performance with lower onset potential of 0.68 V versus RHE and H₂O₂ selectivity (<50%)," It would be difficult to understand why is the H₂O₂ selectivity lower for H₂PcF16 modified electrode (<50%) as compared to that of CoPc and CoPcF16 (>60%).
2. Page 15, lines 385-387. "at 0.63 V versus RHE (equivalent to an overpotential of 230 mV)"; and "at 0.33 V versus 386 RHE (equivalent to an overpotential of 530 mV)" By which means these values of overpotential were calculated? These calculations of overpotential should be provided in Supplementary Information.
3. Supplementary Fig. 25. The unit of the inverse square root of the electrode rotation rate is wrong in Figure 25b,d. It should be rpm instead of rmp.

Typos:

Page 10, line 258. 'at differe- nt sweep rates' Please correct.

Page 15, line 398. 'towards' should be 'towards'

Reviewer #4 (Remarks to the Author):

Regarding previous studies on COFs used in the $2e^-$ oxygen reduction reaction, the authors should at the very least have cited these work, which is arguably relevant, and further that they should have made an attempt to discuss the differences between the references and their work.

Response to reviewers' reports:

Reviewer #2

The current revision could be accepted.

Answer: We really appreciate this reviewer's affirmation.

Reviewer #3

Manuscript: NCOMMS-23-34167B

"Dithiine-linked Metalphthalocyanine Framework with Undulated Layers for Highly Efficient and Stable H₂O₂ Electroproduction", by Qianjun Zhi, Rong Jiang, Xiya Yang, Yucheng Jin, Dongdong Qi, Kang Wang, Yunpeng Liu, and Jianzhuang Jiang

This manuscript could be accepted for publication in Nature Communications after minor revision.

Answer: We truly appreciate this reviewer's affirmation. The responses to the following valuable comments and suggestions towards further enhancing the quality of our research are detailed below.

Minor comments:

1. Page 12, lines 310-312. "However, the metal-free H₂PcF₁₆ electrode displays much inferior electrocatalytic performance with lower onset potential of 0.68 V versus RHE and H₂O₂ selectivity (<50%)," It would be difficult to understand why is the H₂O₂ selectivity lower for H₂PcF₁₆ modified electrode (<50%) as compared to that of CoPc and CoPcF₁₆ (>60%).

Answer: Thanks a lot for this comment! It is worth noting that CoPc-based electrocatalysts have been revealed to show higher selectivity for 2e⁻ ORR compared to 4e⁻ ORR with Co atoms as the catalytic active center, see for examples: *Nat. Commun.* 14, 1426 (2023); *Nat. Commun.* 14, 172, (2023); and *J. Phys. Chem. C* 113, 20689-20697 (2009). In contrast, metal-free H₂Pc-based electrocatalysts usually exhibit very low activity towards both 2e⁻ ORR and 4e⁻ ORR owing to the lack of highly effective active sites, see for examples: *Electroanalysis* 14, 540, (2002) and *J. Electroanal. Chem.* 577, 223-234 (2005). This is also true for metal-free H₂PcF₁₆ in the present case, which exhibits inferior 2e⁻ ORR performance to CoPc and CoPcF₁₆ as certified by the lower onset potential and H₂O₂ selectivity than those for CoPc and CoPcF₁₆, Supplementary Fig. 37 in the revised version of Supplementary Information.

2. Page 15, lines 385-387. "at 0.63 V versus RHE (equivalent to an overpotential of

230 mV)”; and “at 0.33 V versus RHE (equivalent to an overpotential of 530 mV)” By which means these values of overpotential were calculated? These calculations of overpotential should be provided in Supplementary Information.

Answer: Thanks a lot for this comment and suggestion! For electrochemical reaction, the difference between the potential applied during the reaction and the equilibrium potential under the specified reaction conditions (which often differ from standard state conditions) is defined as the “overpotential”, see for examples: *ACS Energy Lett.* **5**, 1083-1087 (2020); *Acc. Chem. Res.* **53**, 561-574 (2020); and *J. Mater. Chem. A* **4**, 17587-17603 (2016). For electrochemical H₂O₂ production in the present work, overpotential can be calculated according to the following equation:

$$\eta = U_0 - U$$

Where η is the overpotential, U is the applied electrode potential, and U₀ is the equilibrium potential of 0.76 V versus RHE at 298 K in an alkaline electrolyte, see for examples: *J. Phys. Chem. B* **108**, 17886-17892 (2004) and *Nat. Catal.* **1**, 282–290 (2018). On the basis of this equation, the applied potentials of 0.33 and 0.63 V versus RHE are corresponding to an overpotential of 430 and 130 mV, respectively, which have been corrected in the revised version of manuscript. According to this suggestion, the calculation method of overpotential has been added into the revised version of Supplementary Information.

3. Supplementary Fig. 25. The unit of the inverse square root of the electrode rotation rate is wrong in Figure 25b,d. It should be rpm instead of rmp.

Answer: Thanks a lot for this comment! Accordingly, ‘rmp’ has been corrected to ‘rpm’ in Figures 25b and 25d in the revised version of Supplementary Information.

Typos:

Page 10, line 258. ‘at differe- nt sweep rates’ Please correct.

Answer: Thanks a lot for this comment! Accordingly, ‘at differe- nt sweep rates’ has been corrected into ‘at different sweep rates’ in the revised version of manuscript.

Page 15, line 398. ‘towards’ should be ‘towards’

Answer: Thanks a lot for this comment! According to the instruction, ‘towards’ has been changed into ‘towards’ in the revised version of manuscript.

Reviewer #4

Regarding previous studies on COFs used in the 2e⁻ oxygen reduction reaction, the authors should at the very least have cited these work, which is arguably relevant, and

further that they should have made an attempt to discuss the differences between the references and their work.

Answer: We indeed appreciate this reviewer's affirmation regarding our work and the valuable suggestions towards further improving the quality of the present research. Accordingly, the reports on the state-of-the-art COF-based electrocatalysts for 2e⁻ ORR have been cited in the revised version of manuscript. It is worth noting that the features of COFs such as the molecularly structural tunability and desired permanent porosity enable this class of materials to be promising advanced catalysts with well-defined structure and active sites for various electrochemical applications. Recently, 2D COFs such as PYTA-TPEDH-COF [*Angew. Chem. Int. Ed.* **62**, e202218742 (2023)]; TP-TD-COF [*Small Struct.* **4**, 2200387 (2023)]; and Py-TD-COF [*Appl. Catal. B.* **340**, 123216 (2024)] as well as 3D COFs including BUCT-COF-1 [*Angew. Chem. Int. Ed.* **62**, e202216751 (2023)] and BUCT-COF-7 [*Angew. Chem. Int. Ed.* e202314539 (2023)] have been employed as metal-free electrocatalysts to promote 2e⁻ ORR with H₂O₂ selectivity up to *ca.* 90%. Introduction of highly efficient Co active sites for 2e⁻ ORR into CoPc-S-COF affords its enhanced H₂O₂ selectivity (> 95%) in the present work. Nevertheless, the undulated layer-stacked structure of CoPc-S-COF favours the exposure of active sites, resulting in the higher H₂O₂ production rate in comparison with all the thus far reported COF-based electrocatalysts, Supplementary Table 8 in the revised version of Supplementary Information. Actually, the electrocatalytic H₂O₂ production performance of CoPc-S-COF is also competitive with that of the state-of-the-art inorganic catalytic materials, Supplementary Table 8 in the revised version of Supplementary Information, demonstrating its great application potential towards practical H₂O₂ production. According to this instruction, corresponding description has been added into the revised version of manuscript.